# WHY ALIGNMENT MUST PRECEDE DISTILLATION: A MINIMAL WORKING EXPLANATION

## ABSTRACT

For efficiency, preference alignment is often performed on compact, knowledge-distilled (KD) models. We argue this common practice introduces a significant limitation by overlooking a key property of the alignment's reference model: its ability to cover the full range of the underlying distribution. We show that the standard KD → Align workflow diminishes the model's capacity to recover specific target capabilities that were pruned during distillation, even under strong preference signals. We instead demonstrate that reversing the pipeline (*i.e.*, Align → KD) is essential: alignment must first be performed on a reference model with broad distributional coverage before distillation. Our contributions are threefold. First, we provide a minimal working explanation of how the reference model constrains preference alignment objectives at a fundamental level. Second, we validate this theory in a controllable Mixture-of-Gaussians experiment, where anchoring to a limited-coverage reference consistently results in suboptimal model performance. Finally, we demonstrate that the same phenomenon holds in LLM alignment with the SmolLM2 family: models aligned after KD fail to effectively recover intended capabilities, resulting in substantially lower reward and target precision. In contrast, our proposed Align → KD pipeline robustly captures these capabilities, yielding models with superior target-oriented metrics and lower variance. Together, these results establish the reference model's distributional coverage as a first-order design choice in alignment, offering a clear principle: *alignment must precede distillation.*

## 1 INTRODUCTION

The alignment of large language models (LLMs) with human preferences has emerged as a central challenge in modern AI research. Building on pretrained models with vast general knowledge, algorithms such as Reinforcement Learning from Human Feedback (RLHF; Ziegler et al. (2019); Stiennon et al. (2020); Ouyang et al. (2022)) via PPO (Schulman et al., 2017) and Direct Preference Optimization (DPO; Rafailov et al. (2023)) have become standard methods. RLHF generally formulates alignment as reward maximization under a Kullback–Leibler (KL) penalty to a fixed reference model, while DPO reparameterizes preference learning into a pairwise loss that still anchors to the same reference. Recent refinements—including GRPO and KTO—improve stability, variance reduction, or gradient calibration, yet all share a structural dependence: alignment is always regularized against a fixed reference model $\pi_{\text{ref}}$ (Shao et al., 2024; Ethayarajh et al., 2024).

The purpose of this anchoring is well-understood. By penalizing divergence from $\pi_{\text{ref}}$, alignment algorithms stabilize optimization, curb drift/forgetting, and confine exploration to plausible regions of the output space (Korbak et al., 2022; Zhang et al., 2025). In RLHF via PPO, reverse KL is used for mode-seeking, while forward KL encourages coverage of $\pi_{\text{ref}}$'s support (Zhang et al., 2025). In DPO, the loss decomposes into a model log-ratio plus a reference log-ratio—a per-example offset: if $\pi_{\text{ref}}$ already ranks correctly, training is easy; if it misranks, gradients diminish and flipping the preference becomes considerably more difficult (Chen et al., 2024). Across methods, $\pi_{\text{ref}}$ functions as the anchor around which preference learning unfolds.

Yet amid this focus on *how* to regularize, a foundational question has been overlooked: *which reference model should we use?* Most works treat the reference $\pi_{\text{ref}}$ as given, optimizing *how* to regularize rather than *which* model to anchor to (Korbak et al., 2022; Zhang et al., 2025). In practice,

this question is often answered implicitly: practitioners frequently employ a *compressed or distilled* checkpoint—often adopting a compact model derived from knowledge distillation (KD) and then using that model as the reference for preference alignment. This choice is driven by pragmatism, as it reduces compute, aligns with the common availability of distilled models, and serves the end goal of a compact final model (Sanh et al., 2019; Tunstall et al., 2023; Dubey et al., 2024; Allal et al., 2025). However, this approach has a significant drawback: KD typically trades coverage for efficiency, pruning rare modes and systematically reducing the model's distributional support (Cha & Cho, 2025).

In this paper, we posit a fundamental coverage condition: target capabilities must lie within the support of $\pi_{\mathrm{ref}}$. This requirement reframes pipeline design. The community's de facto default, **Pipeline K-A (`KD` → `Align`)**, begins from a compact model with reduced coverage, making it vulnerable to a structural failure mechanism we identify as the *low-recall trap*. This failure manifests as two effects: 1) a *sampling trap*, where data collection rarely visits forgotten capabilities, and 2) a *learning trap*, where the very regularization terms designed for stability actively penalize their recovery. Therefore, we advocate **Pipeline A-K (`Align` → `KD`)**: first align a reference with broad coverage, then apply KD. While the intuition that a more capable model aligns more easily is common, our contribution is to formalize this notion as a specific distributional coverage requirement. We make this requirement explicit and validate it empirically: the properties of the anchor—especially its distributional coverage—are a first-order design decision, not an implementation detail.

To substantiate this claim, we adopt a two-stage empirical strategy. First, we introduce a controllable *Mixture-of-Gaussians (MoG) experiment* where distributional coverage can be manipulated directly and alignment dynamics observed precisely. Second, we extend the analysis to LLMs, aligning the `SmolLM2` family under both pipelines. Across settings, the results converge: KD → Align is constrained by limited reference support, while Align → KD produces compact models that remain reliably aligned. This work makes the following contributions:

- We identify reference model coverage as a critical, overlooked factor in preference alignment and provide a minimal working explanation for its impact. Our analysis formalizes this failure mechanism (termed the *low-recall trap*), showing why the standard KD → Align pipeline is structurally flawed, while our proposed Align → KD alternative offers a robust solution.

- In a Mixture-of-Gaussians experiment, we empirically isolate the effect of reduced coverage and provide a precise, experimental analysis of the failure dynamics across alignment algorithms.

- We validate our principle at scale with the `SmolLM2` language model family, demonstrating that reference model coverage is a key determinant of final model performance and training stability in realistic alignment scenarios.

In conclusion, our findings establish a fundamental design principle: *alignment must precede distillation* to ensure both the stability and performance of compact aligned LLMs.

## 2 RELATED WORK

**Preference Alignment of LLMs.** Large-scale alignment commonly follows RLHF with PPO, which anchors the learned policy to a supervised finetuning (SFT) reference via a reverse-KL penalty (Schulman et al., 2017; Ziegler et al., 2019; Stiennon et al., 2020; Ouyang et al., 2022; Bai et al., 2022). Foundational work on learning from preferences predates LLMs (Christiano et al., 2017) and was later adapted to language (Ziegler et al., 2019; Stiennon et al., 2020). Numerous variants—GRPO (Shao et al., 2024), ReMax (Li et al., 2023), RRHF (Yuan et al., 2023)—modify estimators or baselines but keep a fixed reference anchor. DPO removes explicit reward modeling while retaining a reference—centered objective (Rafailov et al., 2023), with practical successors such as KTO (Ethayarajh et al., 2024). Orthogonal to these algorithmic refinements, our work asks *which* model should serve as the anchor.

**Analysis on Preference Alignment Methods.** Most analytical work on preference-based post-training examines objectives, procedures, and outcomes, while taking the properties of the *reference model* as given. Such analyses have clarified the role of KL regularization as a Bayesian prior,

documented trade-offs between RL and SFT (Korbak et al., 2022; Kirk et al., 2024; Shenfeld et al., 2025), and identified biases in reward models or DPO objectives (Gao et al., 2023; Lu et al., 2024). While these threads largely overlook the anchor, its implicit importance is evident in other lines of research. For instance, research on iterative alignment, where a fine-tuned model becomes the anchor for a subsequent stage (Bai et al., 2022; Anil et al., 2023), demonstrates that a stronger reference yields a better final policy, yet does not isolate *why* the new anchor is more effective. Similarly, studies on SFT data quality that emphasize response diversity and coverage (Zhou et al., 2023; Tunstall et al., 2023) highlight the criticality of the initial policy's distribution, but primarily focus on the upstream data rather than the downstream anchor's functional properties for alignment. Our work directly addresses this gap: we isolate the reference model's properties as a first-order design variable, formalize its quality through the lens of distributional coverage, and show this property to be essential in avoiding the low-recall trap.

## 3 THE ROLE OF THE REFERENCE MODEL IN PREFERENCE ALIGNMENT

Modern preference alignment algorithms solve the challenge of steering LLMs toward desired behaviors without catastrophic forgetting (Ouyang et al., 2022) by universally regularizing the learned policy $\pi_\theta$ against a fixed *reference model*, $\pi_{\text{ref}}$ (*i.e.*, a model after supervised fine-tuning). This anchoring stabilizes optimization and prevents destructive updates across algorithmic families (Schulman et al., 2017; Rafailov et al., 2023; Zhang et al., 2025). While this anchoring principle is fundamental to both Reinforcement Learning (RL) and Direct Preference Optimization (DPO), we argue that the field has overlooked a critical question: *which model should serve as the reference in the first place?* We examine this unasked question and its critical consequences for alignment success.

### 3.1 REFERENCE MODELS IN RLHF

RLHF with Proximal Policy Optimization (PPO) has become the standard method for aligning LLMs (Ziegler et al., 2019; Stiennon et al., 2020; Ouyang et al., 2022). Unlike classical PPO, where the reference policy is typically the previous iterate (Schulman et al., 2017), alignment practice almost always anchors to the *initial* supervised fine-tuned (SFT) model as $\pi_{\text{ref}}$. The anchoring is implemented via a Kullback–Leibler (KL) penalty (Ziegler et al., 2019; Ouyang et al., 2022):

$$\mathcal{J}(\theta) = \mathbb{E}_{\pi_\theta}[R(y|x)] - \beta\, D_{\text{KL}}\big(\pi_\theta(y|x) \,\|\, \pi_{\text{ref}}(y|x)\big), \tag{1}$$

where $\beta > 0$ controls the strength of the anchor. The choice of KL direction is consequential: the reverse KL, $D_{\text{KL}}(\pi_\theta \,\|\, \pi_{\text{ref}})$, is *mode-seeking*, concentrating probability where $\pi_{\text{ref}}$ is already confident, while the forward KL, $D_{\text{KL}}(\pi_{\text{ref}} \,\|\, \pi_\theta)$, is *support-covering* (zero-avoiding), encouraging $\pi_\theta$ to cover the support of $\pi_{\text{ref}}$ (Zhang et al., 2025). This is often implemented via reward shaping, $r'(y|x) = R(y|x) - \beta \log \frac{\pi_\theta(y|x)}{\pi_{\text{ref}}(y|x)}$, which yields the same gradient as reverse-KL regularization (Ziegler et al., 2019; Stiennon et al., 2020).[1]

*In practice*, this reverse-KL anchoring to an SFT reference is the de facto standard, used in large-scale deployments such as InstructGPT and Claude (Ziegler et al., 2019; Ouyang et al., 2022; Bai et al., 2022). While subsequent work has proposed numerous refinements to improve stability or reduce variance—such as GRPO (Shao et al., 2024), ReMax (Li et al., 2023), and RRHF (Yuan et al., 2023)—the core mechanism remains unchanged: all variants fundamentally constrain alignment by anchoring to a fixed reference model, $\pi_{\text{ref}}$.

### 3.2 REFERENCE MODELS IN DPO

DPO (Rafailov et al., 2023) removes explicit reward modeling but still places the reference model at the heart of its objective. Rewriting the DPO loss for a preference pair $(y_w, y_l)$ (winner, loser)

---

[1]Forward-KL regularization $D_{\text{KL}}(\pi_{\text{ref}} \| \pi_\theta)$ alleviates the strict support barrier but is rarely used at scale due to instability/variance; moreover, DPO-style objectives retain a reference offset (Sec. 3.4).

reveals an additive decomposition (Chen et al., 2024):

$$\mathcal{L}_{\text{DPO}} \; \propto \; - \, \mathbb{E}_{(y_w, y_l) \sim \mathcal{D}} \left[ \log \sigma \left( \beta \underbrace{\log \frac{\pi_\theta(y_w|x)}{\pi_\theta(y_l|x)}}_{\text{model log-ratio}} + \beta \underbrace{\log \frac{\pi_{\text{ref}}(y_l|x)}{\pi_{\text{ref}}(y_w|x)}}_{\text{reference log-ratio}} \right) \right]. \tag{2}$$

Here, the model log-ratio is what $\pi_\theta$ learns to increase, while the reference log-ratio is a *per-example constant offset* determined entirely by $\pi_{\text{ref}}$. If $\pi_{\text{ref}}$ already prefers the correct candidate, the offset places the sigmoid in its high-slope region, which facilitates the optimization for $\pi_\theta$; conversely, if $\pi_{\text{ref}}$ misranks the pair, the offset shifts the sigmoid toward saturation, diminishing gradients and making it substantially harder to flip the ranking (Chen et al., 2024). Successors like KTO (Etha-yarajh et al., 2024) explore alternative feedback signals but still retain a reliance on the reference model: alignment is anchored, explicitly or implicitly, to $\pi_{\text{ref}}$.

### 3.3 THE UNASKED QUESTION: WHICH REFERENCE MODEL?

Despite the central role of $\pi_{\text{ref}}$, surprisingly little attention has been paid to a more basic question: *which model should serve as the reference?* Prior work has focused almost exclusively on how to formulate the regularization while taking $\pi_{\text{ref}}$ itself as given, typically as the initial SFT check-point (Ouyang et al., 2022). However, a second *de facto* standard has emerged in practice, often followed without deep consideration of its consequences: adopting a smaller, knowledge-distilled (KD) model as the reference. This workflow is motivated by pragmatic concerns such as (i) lower computational costs during alignment, (ii) the goal of producing a compact final model, and (iii) the simple availability of powerful, publicly released KD models (Sanh et al., 2019; Tunstall et al., 2023; Dubey et al., 2024; Allal et al., 2025).

### 3.4 THE LOW-RECALL TRAP: FROM THEORY TO PRACTICE

Using a compact KD model as $\pi_{\text{ref}}$ is not merely suboptimal—it induces a structural failure in the learning dynamics. We define this failure as the *low-recall trap*: a phenomenon where a stabilizing anchor turns into a barrier. This trap arises from the distributional properties of the KD reference and manifests through two compounding stages: a *sampling trap* and a *learning trap*.

**Premise: Distillation Reduces Distributional Recall.** Our analysis rests on the premise that KD inherently reduces distributional recall. Recent work by Cha & Cho (2025) formally justifies this, demonstrating that distillation induces a structural trade-off between precision and recall. Specifi-cally, as the teacher's distribution becomes more selective (*e.g.*, via temperature scaling), the student model minimizes loss by concentrating probability mass on high-density regions at the expense of broader coverage. This mechanism effectively prunes rare modes even under objectives like forward KL, as the student allocates its limited capacity to match the teacher's most emphasized components. Consequently, anchoring alignment to such a *low-recall* reference model creates a critical condition: the probability of generating desirable but rare behaviors $y^\star$ effectively vanishes ($\pi_{\text{ref}}(y^\star|x) \approx 0$).

**Sampling Trap.** This probability collapse creates the first barrier during data generation. In on-policy alignment, on-policy data are generated by $\pi_\theta$, but a large reverse-KL penalty keeps $\pi_\theta$ close to $\pi_{\text{ref}}$; in early/mid training the effective sampling distribution remains confined to high-probability regions of $\pi_{\text{ref}}$. If a *desirable behavior* $y^\star$ was pruned during distillation, then $\pi_{\text{ref}}(y^\star|x) \approx 0$ and $\pi_\theta(y^\star|x)$ stays negligible, making it unlikely that the required examples ever enter the dataset. Note that this sampling trap extends to offline alignment (*e.g.*, DPO), as preference datasets collected from a low-recall $\pi_{\text{ref}}$ inherit the same lack of coverage.

**Learning Trap in PPO.** The second barrier arises during optimization. In PPO, the policy update is driven by a reward signal shaped by the KL penalty: $r'(y|x) = R(y|x) - \beta \log(\pi_\theta(y|x)/\pi_{\text{ref}}(y|x))$. For a desirable response $y^\star$ with $\pi_{\text{ref}}(y^\star|x) \approx 0$, the KL penalty term explodes, even if the reward model assigns a high reward $R(y^\star|x)$:

$$\lim_{\pi_{\text{ref}}(y^\star|x) \to 0} \left( -\beta \log \frac{\pi_\theta(y^\star|x)}{\pi_{\text{ref}}(y^\star|x)} \right) = -\infty. \tag{3}$$

The shaped reward becomes infinitely negative, overwhelming any positive signal; exploratory moves toward $y^\star$ are penalized, effectively trapping the policy within the limited support of the

low-recall reference. Moreover, RLHF variants such as GRPO, ReMax, and RRHF differ mainly in baselines or estimators but retain reverse-KL anchoring or equivalent shaping, so the same low-recall mechanism persists.

**Learning Trap in DPO.** Let $z := \beta \left( \log \frac{\pi_\theta(y_w|x)}{\pi_\theta(y_l|x)} + \log \frac{\pi_{\text{ref}}(y_l|x)}{\pi_{\text{ref}}(y_w|x)} \right)$ be the logit in equation 2. The per-pair loss is $-\log \sigma(z)$ and

$$\frac{\partial \mathcal{L}_{\text{DPO}}}{\partial \Delta_\theta} = -\beta \left( 1 - \sigma(z) \right) = -\beta \, \sigma(-z), \qquad \Delta_\theta := \log \frac{\pi_\theta(y_w|x)}{\pi_\theta(y_l|x)}.$$

If $\pi_{\text{ref}}(y_w|x) \approx \varepsilon \ll 1$ while $\pi_{\text{ref}}(y_l|x)$ is moderate, the reference offset makes $z \gg 0$ even when $\Delta_\theta \approx 0$, so $\sigma(-z) \approx 0$ and the gradient vanishes. Thus, pairs involving missing/rare modes receive negligible updates. This offset-induced saturation persists in DPO-style objectives like KTO (Ethayarajh et al., 2024), whenever a fixed low-recall reference is retained.

**From Theory to an Empirical Question.** While this analysis illustrates a catastrophic failure in the limit where $\pi_{\text{ref}}(y^\star|x) \approx 0$, real-world scenarios may be less extreme; probabilities for desirable behaviors, while low, are rarely identically zero. Nonetheless, the core issue persists: KD is known to systematically degrade recall (Cha & Cho, 2025), causing the probabilities of desirable behaviors, $\pi_{\text{ref}}(y^\star|x)$, to become exceedingly small. For any practical value of $\beta$, the resulting reverse-KL penalties and DPO offsets can still grow large enough to overwhelm the preference signal, preserving the fundamental trap. This leads to an empirical question: *do the failures predicted by our analysis occur in practice, even when evaluated with an ideal reward oracle?*

### 3.5 Pipeline Choice as a First-Order Design Decision

Our analysis thus shifts the focus from *how* to regularize to a more fundamental question of *what* to anchor to. This reframes the challenge as a critical pipeline choice between two distinct strategies:

- **Pipeline K-A (`KD` → `Align`)**: The default workflow, which anchors alignment to a compact but low-recall reference model, triggering the sampling and learning traps.
- **Pipeline A-K (`Align` → `KD`)**: Our proposed workflow, which first aligns a high-recall reference model to satisfy preference constraints and then distills it into a compact model.

While it may sound intuitive that aligning a larger, more capable model is preferable, our contribution is to move beyond this intuition. We provide the first rigorous and generic validation of this principle, analyzing the precision-recall trade-offs in both a fully controllable synthetic environment and LLM experiments.

## 4 Experimental Validation

We now empirically validate our central claim: anchoring alignment to a low-recall reference induces systematic failure. We analyze this failure through the lens of precision and recall, demonstrating that preserving the reference model's recall is essential for achieving robust alignment. We adopt a two-stage methodology: first, a controllable *Two-Dimensional Mixture-of-Gaussians (MoG) experiment* that isolates the dynamics of the low-recall trap; second, *LLM validation* with the `SmolLM2` family to verify that the same failure mode persists in realistic pipelines.

### 4.1 Empirical Validation with Mixture-of-Gaussians

**Experimental Setup.** The MoG experiment lets us precisely manipulate recall. We define a ground-truth distribution $p^*$ over a 2D Euclidean space ($\mathbb{R}^2$), consisting of 8 modes, a high-recall model $p'$ (6 modes), and two low-recall KD models $p_3''$ (3 modes) and $p_4''$ (4 modes). We designate one of the 8 modes from $p^*$ as the alignment target, and define the target distribution, $p_{\text{target}}$, as the Gaussian distribution of this single mode. We compare two pipelines: **Pipeline K-A (`KD` → `Align`)**, which uses the low-recall $p''$ as the reference $\pi_{\text{ref}}$, and **Pipeline A-K (`Align` → `KD`)**, which uses the high-recall $p'$. Following Cha & Cho (2025), we obtain $p''$ from $p'$ by reparameterizing the mixture weights with a temperature-like parameter $\beta_{\text{KD}} \geq 1$. We provide further details on the experimental setup and hyperparameters in App. A.

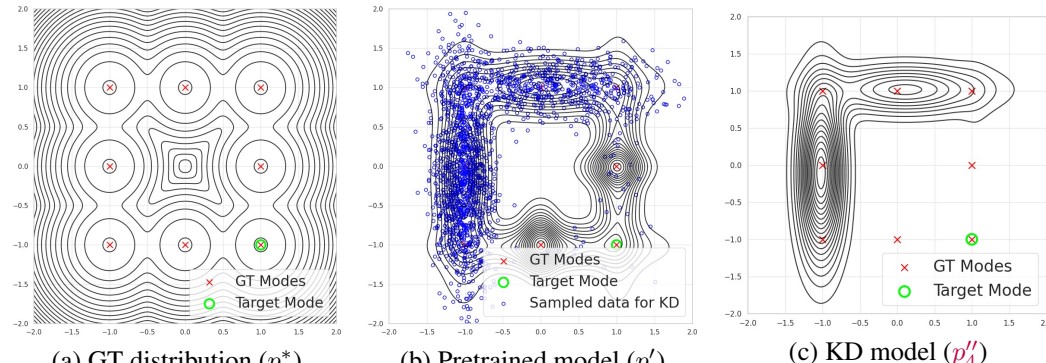

(a) GT distribution ($p^*$)     (b) Pretrained model ($p'$)     (c) KD model ($p_4''$)

Figure 1: Mixture-of-Gaussians experiment. (a) Ground-truth $p^*$ with eight modes. (b) High-recall $p'$ (six modes fit to samples from $p^*$): Overall Precision $= -2.2720$, Overall Recall $= -2.0054$, Target Precision $= -34.7812$. (c) Low-recall $p_4''$ (distilled from $p'$ using $\beta_{\mathrm{KD}} = 10$): Overall Precision $= -2.2703$, Overall Recall $= -2.9604$, Target Precision $= -51.4754$. Green circle denotes the target distribution of the single mode. Note that distillation may drop rare modes, reducing recall and target precision.

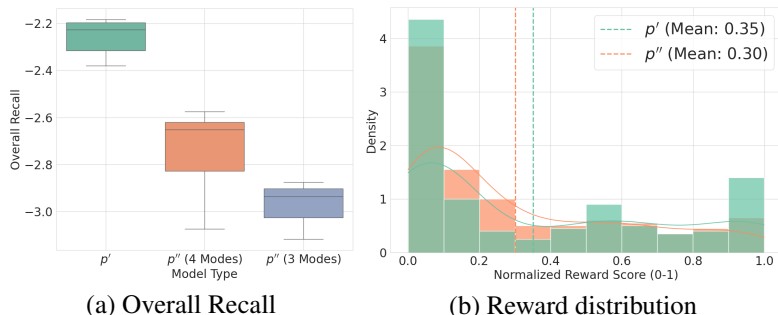

(a) Overall Recall        (b) Reward distribution

Figure 2: Learning and sampling trap check. (a) $p'$ (Pipeline A-K reference) consistently attains higher Overall Recall than the KD model $p_3''$ (Pipeline K-A reference). (b) Samples from $p'$ populate the high-reward region under the oracle substantially more often, evidencing a *sampling trap* for the KD model $p_3''$.

**Oracle Reward Function.** To quantify the alignment of generated 2D samples toward the target mode, we design a simple and deterministic oracle reward $R(\mathbf{x})$ defined by the squared Euclidean distance to the target center $\mathbf{c}_t$:

$$R(\mathbf{x}) = 10.0 \cdot \exp\left(-\alpha \cdot ||\mathbf{x} - \mathbf{c}_t||^2\right), \tag{4}$$

where $\alpha = 2.0$ controls the sharpness. This yields a dense reward signal ranging from 0 to 10.0, peaking at the target.

**Evaluation Metrics.** To assess outcomes, we adapt the precision–recall framework of Cha & Cho (2025), reporting four complementary metrics for a final model $q$:

$$\text{Overall Precision} := \mathbb{E}_{x \sim q}\big[\log p^*(x)\big], \qquad \text{Target Precision} := \mathbb{E}_{x \sim q}\big[\log p_{\text{target}}(x)\big], \tag{5}$$

$$\text{Overall Recall} := \mathbb{E}_{x \sim p^*}\big[\log q(x)\big], \qquad \text{Final Average Reward} := \mathbb{E}_{x \sim q}\big[R(x)\big]. \tag{6}$$

Overall Precision evaluates whether samples from $q$ are plausible under the ground-truth $p^*$; Overall Recall measures $q$'s coverage of $p^*$. Target Precision and Final Average Reward quantify concentration on the rewarded subset $p_{\text{target}}$ and on the reward function $R(x)$, respectively—disentangling the dual goals of alignment: focus on the target while preserving broad coverage.

Building on these metrics, Figure 1 shows that temperature-based distillation from $p'$ produces a more peaked $p''$ that loses several rare modes, including one in $p_{\text{target}}$, sharply decreasing Overall Recall and Target Precision. This creates a challenging low-recall starting point for Pipeline K-A.

**Learning and Sampling Trap Check.** Figure 2 quantifies the initialization gap between the teacher $p'$ (Pipeline A-K reference) and the KD student $p_3''$ (Pipeline K-A reference). First, $p'$ con-

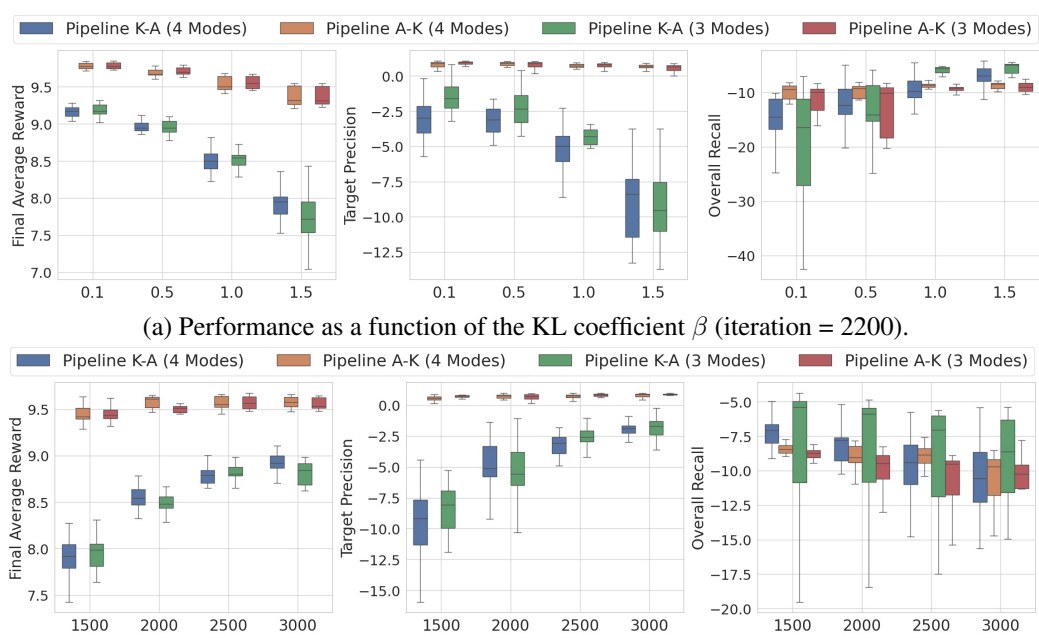

(a) Performance as a function of the KL coefficient $\beta$ (iteration = 2200).

(b) Performance as a function of the number of training iterations ($\beta = 1.0$).

Figure 3: RL (PPO) experiments comparing Pipeline K-A (yielding $p''_{\text{KA}}$) and Pipeline A-K (yielding $p''_{\text{AK}}$). The boxplots summarize results over 20 seeds, sweeping across (a) the KL coefficient $\beta$ and (b) the number of training iterations. We report three key metrics: Final Average Reward, Target Precision, and Overall Recall. Across all conditions, Pipeline A-K consistently achieves superior target-oriented metrics and rewards with significantly lower variance.

sistently exhibits higher Overall Recall than $p''_3$, indicating that KD prunes low-mass modes. This low recall is precisely what sets up a *learning trap*: under reverse-KL shaping (PPO) or a large reference log-ratio (DPO), updates that would recover those pruned modes are discouraged even if they are desirable. Second, while the oracle-reward densities in Figure 2 (b) appear broadly similar, a closer inspection reveals critical differences: $p'$ not only has a slightly higher mean reward but, more importantly, produces more than double the number of samples in the maximum-reward region (normalized reward $\approx 1$) compared to $p''_3$. This tail asymmetry is a clear *sampling trap*: the low-recall reference provides fewer opportunities to observe target-consistent trajectories in the first place. Importantly, these starting differences are modest; the key question is whether alignment attenuates or amplifies them. As we show next (Figs. 3 and 4), training amplifies these gaps into pronounced performance divergences across preference alignment pipelines.

**Results with RL (PPO).** Figure 3 presents the results of our RLHF experiments, systematically comparing Pipeline K-A and Pipeline A-K across different KL coefficients ($\beta$) and training iterations. The findings reveal a clear and consistent pattern: *Pipeline A-K robustly outperforms Pipeline K-A in both target mode alignment and stability*. Across nearly all settings, Pipeline A-K achieves significantly higher Final Average Reward and Target Precision. Furthermore, its markedly lower variance across the 20 trials, visible in the tighter box plot distributions, demonstrates that it is a substantially more stable and reliable alignment process.

As shown in Figure 3 (a), this performance gap is particularly revealing when analyzing the effect of the KL coefficient, $\beta$. Under moderate regularization ($\beta \leq 0.5$), Pipeline A-K successfully acquires the target behavior while achieving a high mean recall, whereas Pipeline K-A often plateaus early with poor target concentration. At large $\beta$ values ($\geq 1.0$), Pipeline K-A sometimes achieves a higher mean *Overall Recall*, but this proves to be "misleading recall": it is accompanied by a collapse in *Target Precision*, indicating that recall is gained by spreading probability mass indiscriminately rather than by recovering the forgotten target mode.

The inferiority of Pipeline K-A is fundamental and could not be remedied by simply increasing the optimization budget or applying stricter compression. Figure 3 (b) shows increasing the iteration budget did not resolve its failure, as its reward curves saturated quickly, whereas Pipeline A-K achieved high rewards even with few iterations. This suggests the bottleneck is the initial model's

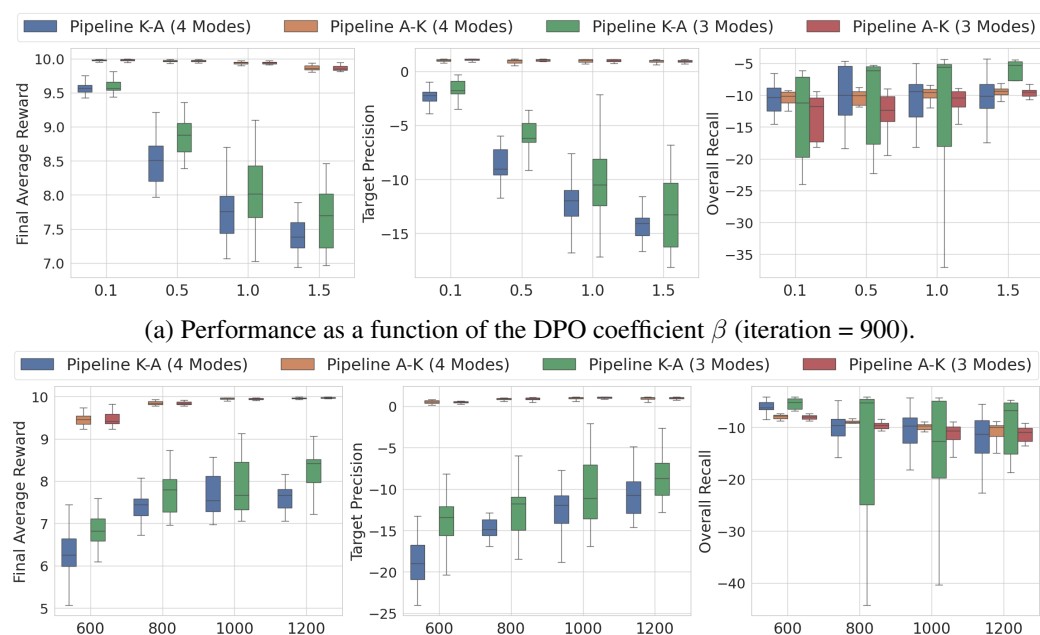

(a) Performance as a function of the DPO coefficient $\beta$ (iteration = 900).

(b) Performance as a function of the number of training iterations ($\beta = 1.0$).

Figure 4: On-policy DPO experiments comparing Pipeline K-A (yielding $p''_{KA}$) and Pipeline A-K (yielding $p''_{AK}$). The boxplots summarize results over 20 seeds, sweeping across (a) the coefficient $\beta$ and (b) the number of training iterations. We report three key metrics: Final Average Reward, Target Precision, and Overall Recall. Consistent with the RL (PPO) findings, Pipeline A-K achieves superior target-oriented metrics and rewards with significantly lower variance.

coverage, not the training budget. Moreover, under a stricter 3-mode constraint, Pipeline K-A's instability was exacerbated, with high variance and frequent target loss across seeds, while Pipeline A-K remained stable. This highlights that preserving recall before alignment is especially critical when the final model must be highly compact. Finally, these trends are mirrored in our GRPO experiments (App. B.1).

**Results with DPO.** The failure of the low-recall pipeline is not an artifact of PPO; the same dynamic emerges under DPO. To ensure a fair comparison with RL's on-policy nature and to isolate the effect of the reference model, Figure 4 presents results from an *on-policy* DPO variant where fresh samples are drawn each iteration and evaluated against a perfect preference oracle.

DPO reproduces the same pattern observed with PPO. Across matched $\beta$ and iteration sweeps, *Pipeline A-K is consistently superior and more stable*, while Pipeline K-A underperforms on Final Average Reward and Target Precision and exhibits higher variance. Furthermore, our off-policy DPO experiments in App. B.2 confirms the stability and superiority of Pipeline A-K.

**Overall Precision Results.** We analyze the Overall Precision to evaluate whether the final models generate samples consistent with the ground-truth distribution. The results, which hold across all tested algorithms, show that *Pipeline A-K consistently achieves not only higher mean Overall Precision but also lower variance* compared to Pipeline K-A. Detailed results for PPO, GRPO, and DPO are presented in App. B.3.

We further extended our analysis to a multiple target mode setting (*i.e.*, targeting both the original mode and its immediate left neighbor in Figure 1(c)). As detailed in Appendix B.4, these experiments yielded trends consistent with the single target mode results. Collectively, the MoG experiments demonstrate that the failure of Pipeline K-A is a structural consequence of the reference model's limited coverage (validating the *low-recall trap*), independent of the specific alignment algorithm or sampling protocol. In contrast, our results confirm that Pipeline A-K avoids this trap, consistently achieving superior outcomes. Building on these synthetic insights, the next section investigates whether this phenomenon persists in large-scale language model alignment.

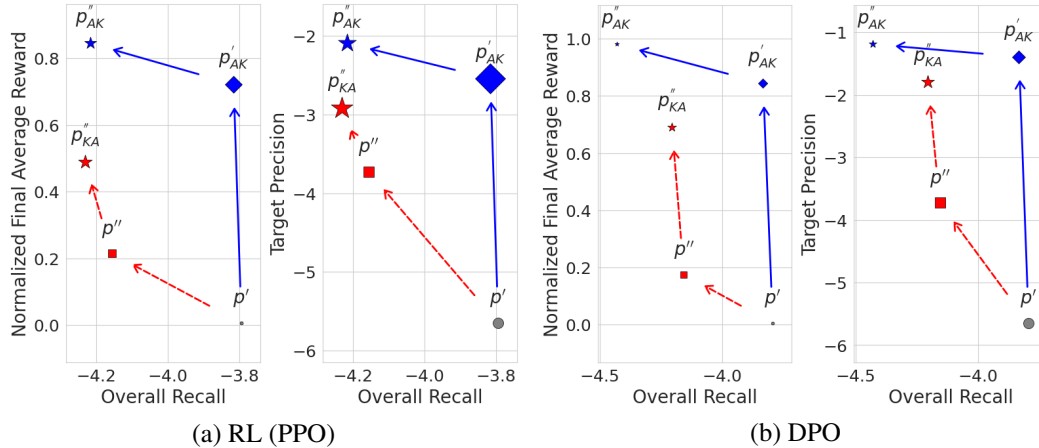

(a) RL (PPO)  (b) DPO

Figure 5: LLM alignment trajectories comparing Pipeline K-A (red) and Pipeline A-K (blue). The plots show the evolution of models for (a) RLHF (PPO) and (b) DPO in the performance space defined by Overall Recall (x-axis) and target-oriented metrics (y-axis). Each marker indicates the mean performance over three seeds, with its size proportional to the cross-seed standard deviation (instability). Arrows depict the pipeline evolution. Consistent with our MoG findings, both algorithms show that Pipeline A-K follows a robust trajectory to a superior and more stable final model.

## 4.2 LLM VALIDATION WITH THE SMOLLM2 FAMILY

While the previous section focused on synthetic Gaussian mixtures, autoregressive LMs are essentially infinite mixtures, where each token distribution acts as a mixture component (Cha & Cho, 2025). Moreover, because the expressivity of each token distribution is bounded by hidden-state dimensionality (Yang et al., 2018), smaller models inherently cover fewer modes, mirroring the bottlenecks observed in our MoG setup. This structural parallel directly connects our MoG analysis to LLMs and motivates the validation experiments that follow.

**Experimental Setup.** To validate our principle in a realistic setting, we use the `SmolLM2` family, adapting the multi-stage setup of Cha & Cho (2025). We treat the pretrained `SmolLM2-1.7B` as the ground-truth distribution ($p^*$). From this, we sample a dataset (temperature $\tau = 1.0$) to train a `SmolLM2-360M`, which serves as our high-recall model ($p'$), which acts as the reference for Pipeline A-K. Subsequently, we distill $p'$ at $\tau = 0.95$ to create our low-recall KD model, a `SmolLM2-135M` ($p''$), which is used as the reference for Pipeline K-A. For all sampling, we use the simple prompt "`The`" to generate sentences. Let $p''_{\text{KA}}$ denote the final model from Pipeline K-A. In Pipeline A-K, let $p'_{\text{AK}}$ be the intermediate model after alignment (aligning from the high-recall $p'$), and $p''_{\text{AK}}$ be the final model after distillation. All experiments use the TRL library (von Werra et al., 2020), with results averaged over three seeds (details in App. C)

**Oracle Reward Function.** To define a target and a reward oracle, we distill $p'$ again at a low temperature ($\tau = 0.8$) to train another `SmolLM2-135M`, denoted $p_{\text{target}}$. Since low-temperature distillation yields a policy concentrated on high-probability modes (*i.e.*, high precision, low recall), $p_{\text{target}}$ serves as an effective oracle for our alignment task (Cha & Cho, 2025). This allows us to design a reward (or preference) function based on the Negative Log-Likelihood (NLL) of a sentence under $p_{\text{target}}$. Detailed formulation and implementation specifics are provided in App. C.3.

**Model Selection.** A critical challenge in reward-maximizing alignment is *mode collapse*, where the policy converges to generating a few high-reward sequences, thereby sacrificing output diversity (Gao et al., 2023; Kirk et al., 2024). Simply selecting the model checkpoint with the highest final reward can lead to this suboptimal outcome. To address this, we employed an early stopping strategy based on criteria that balance reward maximization with behavioral diversity. A detailed description of our model selection protocol, including performance trajectories, is provided in App. C.4.

**Results.** Figure 5 presents our LLM validation using the `SmolLM2` family with PPO and on-policy DPO. The PPO experiments provide clear confirmation of our principle. Crucially, our model selection protocol ensures that both final models, $p''_{\text{AK}}$ and $p''_{\text{KA}}$, were chosen based on criteria that prevent mode collapse and promote response diversity. Even under this diversity-controlled comparison, *the final model from Pipeline A-K ($p''_{\text{AK}}$) outperforms its counterpart ($p''_{\text{KA}}$) across all*

*metrics*—Final Average Reward, Target Precision, and Overall Recall—while also showing better stability. Furthermore, even the intermediate high-recall model ($p'_{AK}$) already surpasses the final aligned low-recall model ($p''_{KA}$) in reward and precision, confirming the severity of the low-recall trap.

The DPO experiments further confirm the superiority of Pipeline A-K in this LLM setting, revealing a similar, albeit more nuanced, pattern where $p''_{AK}$ again achieves superior reward and precision with lower variance. While $p''_{KA}$ exhibits marginally higher Overall Recall in this case, this is not a failure of our pipeline. Instead, it highlights a key feature of the Align $\rightarrow$ KD approach: the final distillation step introduces a predictable, tunable precision-recall trade-off. As established by Cha & Cho (2025), distillation naturally optimizes for precision, which may inherently reduce overall recall. Crucially, however, this trade-off is controllable via the distillation temperature. As demonstrated in Figure 15 (using $\tau = 0.925$ for PPO and $\tau = 0.975$ for DPO), adjusting $\tau$ allows $p''_{AK}$ to effectively balance precision and recall. Comprehensive details are provided in App. D.

In conclusion, these experimental results with LLMs elevate Pipeline A-K from a simple high-performance method to a flexible framework, empowering practitioners to tune compact models according to their specific alignment goals.

## 5 CONCLUDING REMARKS

The prevailing practice in building efficient, aligned language models is to distill a large model into a smaller one before applying costly preference alignment. Our findings challenge this workflow. Across both a Mixture-of-Gaussians experiment and LLM experiments, we show that the distill-first approach introduces a structural *low-recall trap* that constrains alignment to suboptimal outcomes. This trap emerges because alignment amplifies even small differences in reference-model recall, sometimes producing a misleading recall effect: overall recall appears high, yet precision on rare but desirable behaviors collapses. The robust and more efficient alternative is to reverse the pipeline: *alignment must precede distillation*. By first aligning a high-recall reference model and only then distilling its capabilities, one can obtain compact models with higher rewards, stronger target precision, and more stable training dynamics.

These results establish reference-model recall as a first-order design parameter. Beyond challenging current practice, they underscore that pipeline design directly determines the reliability and efficiency of preference alignment, with important implications for scaling aligned language models in both research and deployment.

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

# A MoG Experiment Details

## A.1 Common Experimental Setup

Our Mixture-of-Gaussians (MoG) experiments are designed to simulate the alignment and distillation of language models in a controlled 2D environment. The **ground-truth (GT) distribution** is a uniform mixture of 8 isotropic Gaussian modes, each with a covariance of $0.05 \times \mathbf{I}$, arranged in a $3 \times 3$ grid with the center missing. The **target behavior** is defined as recovering one specific mode (mode #7 located at $[1.5, -1.5]$). All models were implemented as `MoGModel` classes in PyTorch, and all experiments were conducted for `N_TRIALS = 20` independent runs per setting to ensure statistical robustness.

From this GT distribution, we create two types of reference models to initialize our alignment pipelines:

- **High-Recall Model** ($p'$): This model is created by supervised fine-tuning (SFT) a 6-mode MoG model (`N_SFT_MODES = 6`) on samples drawn from the 8 GT modes. Training is conducted for `N_ITERATIONS_SFT_KD = 2000` iterations. This model represents a broad, pre-trained model with high recall of general behaviors but a lack of specific alignment.

- **Low-Recall Model** ($p''$): This model is generated by distilling the high-recall model ($p'$) into a more compact model with fewer components (`N_FINAL_MODES` of 4 or 3). The process uses knowledge distillation (KD). To control the entropy of the teacher distribution during sampling, we reparameterize its mixture weights $\alpha'_k$ using a temperature-like parameter $\beta_{\text{KD}} \geq 1$ (Cha & Cho, 2025):

$$\alpha'_k(\beta_{\text{KD}}) = \frac{\exp(\beta_{\text{KD}} \log \alpha'_k)}{\sum_{j=1}^{K'} \exp(\beta_{\text{KD}} \log \alpha'_j)}. \tag{7}$$

  As $\beta_{\text{KD}}$ increases, the teacher's sampling distribution becomes more peaked, concentrating probability mass on its dominant modes. For our experiments, we use a value of $\beta_{\text{KD}} = 1.25$ (referred to as `KD_SAMPLING_BETA` in our codebase). This model, also trained for `N_ITERATIONS_SFT_KD = 2000` iterations, represents a compact model that has lost some behavioral modes (lower recall) due to distillation.

## A.2 Algorithm-Specific Configurations

All alignment algorithms were trained with a learning rate of `1e-2` and a batch size of `256`. The final distillation step in **Pipeline A-K** (from the aligned high-recall model $p'_{\text{AK}}$ to the final compact model $p''_{\text{AK}}$) consistently used the same KD hyperparameters as those used to create the initial low-recall model.

**Reward Formulation for MoG Experiments.** For the MoG experiments, we use a deterministic, oracle reward function. For a given sample $\mathbf{x}$, the reward $R(\mathbf{x})$ is calculated based on its squared Euclidean distance to the target mode's center, $\mathbf{c}_t$:

$$R(\mathbf{x}) = 10.0 \cdot \exp\left(-\alpha \cdot ||\mathbf{x} - \mathbf{c}_t||^2\right), \tag{8}$$

where the scaling factor $\alpha = 2.0$ controls the sharpness of the reward peak. This function provides a dense reward signal that is higher for samples closer to the target.

**Preference Generation for DPO.** For DPO, which learns from preferences, oracle preference pairs $(y_w, y_l)$ are generated from the reward functions described above. For any two sampled responses $y_1$ and $y_2$ given the same prompt, the response yielding the higher reward ($R(\mathbf{x})$ in the MoG experiments).

**PPO and GRPO (RLHF).** While sharing the same goal of policy optimization, our PPO and GRPO implementations differ fundamentally in their approach to variance reduction and policy updates. PPO utilizes a standard actor-critic framework. It trains a critic network (`ValueModel`) alongside the policy to learn a state-dependent baseline, $V(s)$. This learned baseline is used to

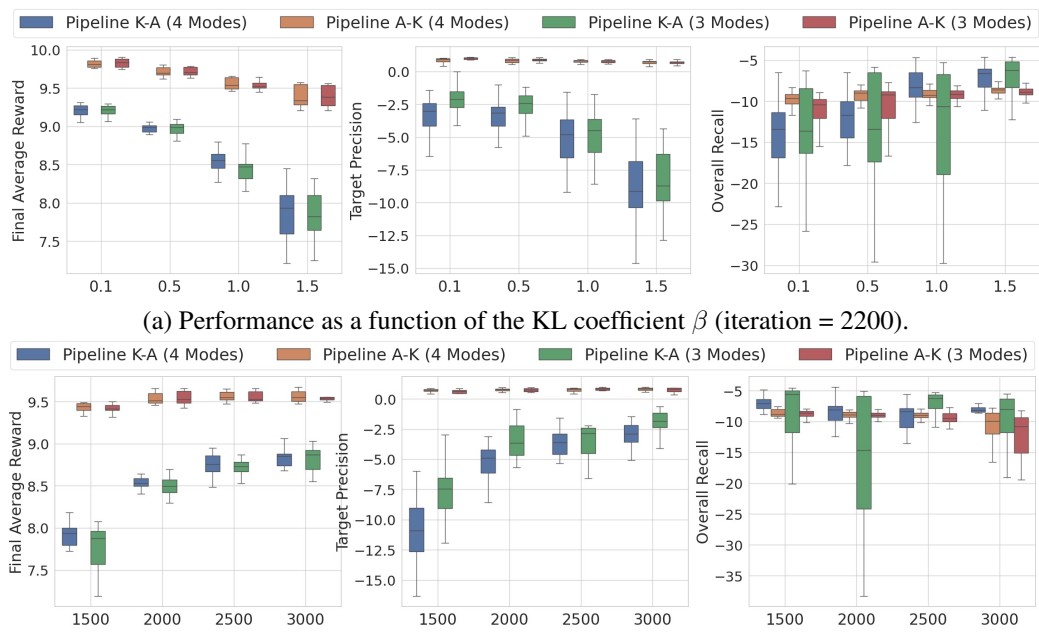

(a) Performance as a function of the KL coefficient $\beta$ (iteration = 2200).

(b) Performance as a function of the number of training iterations ($\beta = 1.0$).

Figure 6: RL (GRPO) experiments comparing Pipeline K-A and Pipeline A-K. The boxplots summarize results over 20 seeds, sweeping across (a) the KL coefficient $\beta$ and (b) the number of training iterations. We report three key metrics: Final Average Reward, Target Precision, and Overall Recall. Across all conditions, Pipeline A-K consistently achieves superior target-oriented metrics and rewards with significantly lower variance. In contrast, Pipeline K-A is unstable and often fails to improve target coverage, confirming the superiority of the `Align → KD` approach.

compute a sophisticated advantage function ($A(s, a) = R(s, a) - V(s)$), which effectively reduces the variance of the policy gradient.

In contrast, our GRPO implementation is critic-free. To reduce variance, it employs a simpler but computationally lighter baseline: the **mean reward of the samples within each batch**. The advantage is calculated as the difference between an individual sample's reward and this batch-mean-reward. This advantage is then used to perform a more direct policy gradient update, which is regularized by a KL divergence penalty.

**DPO.** Our DPO (Rafailov et al., 2023) implementation is critic-free and learns directly from preference pairs. To thoroughly test its robustness, we implemented and experimented with both on-policy and off-policy versions. In the **on-policy** setting, preference pairs are generated on-the-fly from the current policy at each training step. In the **off-policy** setting, a static dataset of preference pairs is generated once from the initial reference model, and the policy is trained over this fixed dataset.

# B  ADDITIONAL EXPERIMENTAL RESULTS WITH THE MOG

## B.1  GRPO

We repeated our analysis using GRPO (Shao et al., 2024), a direct policy gradient algorithm that uses a batch-mean-reward baseline instead of a learned critic. The results, presented in Figure 6, closely parallel those from our PPO experiments. The findings confirm that the superiority of Pipeline A-K is not specific to a single algorithm. Across sweeps of both the KL coefficient and the number of training iterations, **Pipeline A-K consistently achieves higher Final Average Reward and Target Precision with markedly lower variance**. While Pipeline K-A shows moments of high Overall Recall under certain hyperparameters, it does so with significant instability and a frequent collapse in target-oriented metrics. These results reinforce our central conclusion that for stable and effec-

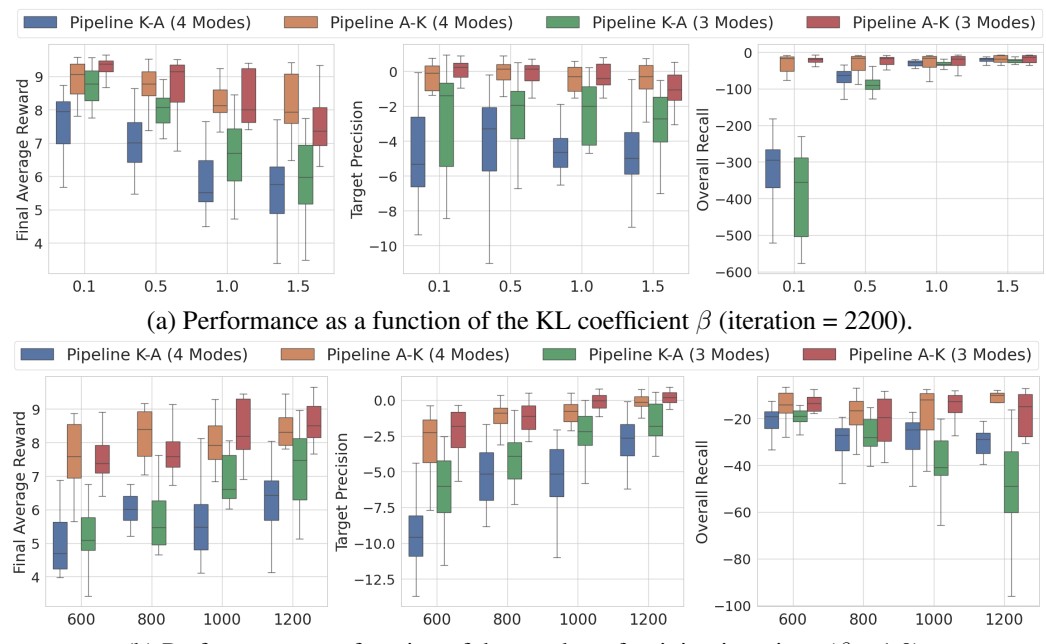

(a) Performance as a function of the KL coefficient $\beta$ (iteration = 2200).

(b) Performance as a function of the number of training iterations ($\beta = 1.0$).

Figure 7: Off-policy DPO experiments comparing Pipeline K-A and Pipeline A-K. The boxplots summarize results over 20 seeds, sweeping across (a) the KL coefficient $\beta$ and (b) the number of training iterations. Pipeline A-K consistently achieves superior target-oriented metrics and rewards. In contrast to on-policy results, both pipelines exhibit comparable variance, while Pipeline A-K also maintains a distinct advantage in Overall Recall.

tive alignment, the choice of a high-recall reference model is critical, regardless of the specific RL algorithm used.

### B.2 OFF-POLICY DPO

We further validate our findings using Direct Preference Optimization (DPO), a critic-free algorithm that learns directly from preference pairs. To test the robustness of our conclusions, we experimented with both on-policy DPO (results in the manuscript) and off-policy DPO, with the off-policy results presented here in Figure 7.

The off-policy DPO results largely corroborate our primary findings. Consistent with the on-policy experiments, **Pipeline A-K demonstrates superior performance in Final Average Reward and Target Precision** across nearly all hyperparameter settings. However, we observe two notable differences from the on-policy case.

First, the performance variance of the two pipelines becomes much more comparable. In the on-policy setting, Pipeline A-K was exceptionally stable, while Pipeline K-A exhibited high variance. In the off-policy setup, however, Pipeline A-K's variance increases, resulting in the two pipelines exhibiting much more comparable stability. We hypothesize this is a direct result of the static training data. The fixed preference dataset provides a more consistent learning signal for the poorly-initialized Pipeline K-A, mitigating the instabilities seen during on-policy exploration. Conversely, for the well-initialized Pipeline A-K, optimizing over a fixed, potentially less diverse dataset may present a noisier optimization landscape, thus slightly increasing its variance.

Second, Pipeline A-K achieves consistently superior Overall Recall across all tested conditions, a significant departure from the "misleading recall" phenomenon. We attribute this to the synergy between a high-recall starting point and the nature of off-policy learning. Pipeline A-K begins with a model that already covers a broad range of behaviors. When trained on a fixed preference dataset, it can effectively shift probability mass to the target mode without the exploratory pressure that might lead to forgetting other modes. In contrast, Pipeline K-A starts with fewer modes and cannot easily "invent" new ones from a static dataset, thus failing to match the recall of a better-initialized model.

## B.3 OVERALL PRECISION RESULTS

To complement the analysis in the manuscript, we report the *Overall Precision* results for the MoG experiments across all four alignment algorithms: PPO, GRPO, on-policy DPO, and off-policy DPO. Overall Precision measures the expected log-likelihood of samples from the final aligned model under the ground-truth distribution $p^*$, thus quantifying the general plausibility of the generated outputs. The results are presented in Figure 8 and Figure 9.

The findings are highly consistent with our other reported metrics. Across all four algorithms and nearly all hyperparameter settings, **Pipeline A-K demonstrates a clear advantage in Overall Precision**. As shown in the figures, the final models from Pipeline A-K ($p''_{AK}$) consistently achieve a higher mean precision than those from Pipeline K-A ($p''_{KA}$). Furthermore, Pipeline A-K exhibits markedly lower variance across the 20 seeds, indicating a more stable and reliable outcome in terms of output plausibility.

This result provides additional evidence against the KD $\rightarrow$ Align approach. Even when Pipeline K-A manages to align toward the target mode (as seen in the manuscript), it often does so at the cost of distorting the overall distribution, leading to less plausible samples. In contrast, the Align $\rightarrow$ KD workflow, by first aligning a high-coverage model and then carefully compressing it, is more effective at preserving the underlying structure of the true data distribution.

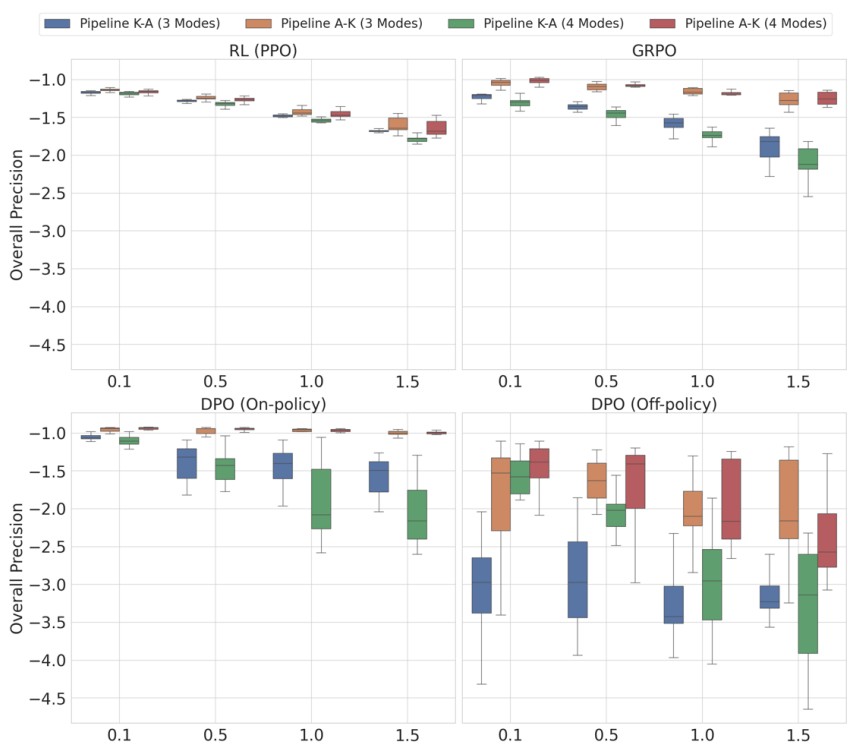

Figure 8: **Overall Precision as a function of the KL coefficient** $\beta$. Each subplot contains a result for each algorithm, comparing Pipeline K-A and Pipeline A-K. Pipeline A-K consistently achieves higher precision with lower variance.

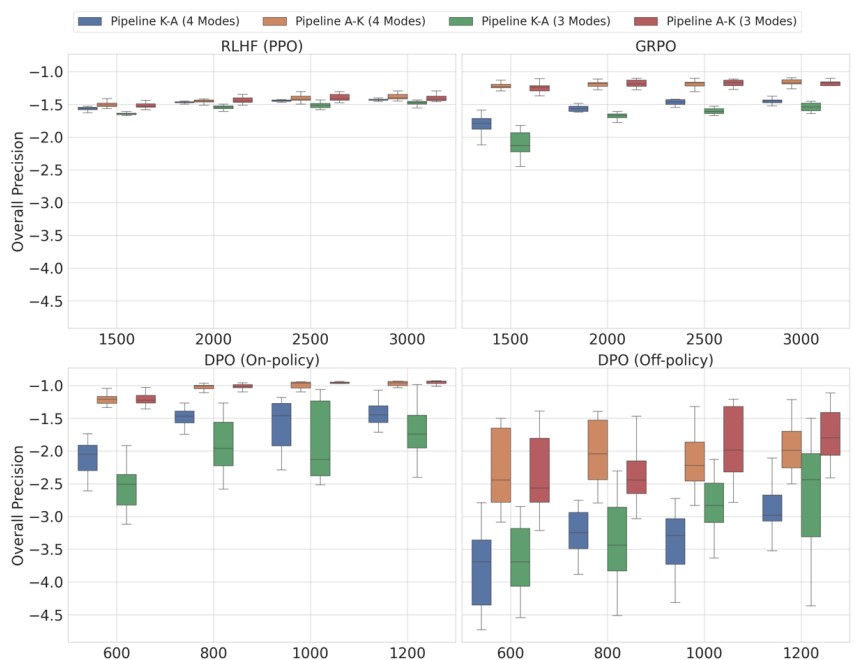

Figure 9: Overall Precision as a function of the number of training iterations. The superiority of Pipeline A-K in both mean performance and stability is consistent across the training process.

### B.4 EXPERIMENTAL RESULTS WITH MULTIPLE TARGET MODES

We present the results of the multi-mode experiments in Figure 10 (PPO), Figure 11 (DPO) and Figure 12 (GRPO). The findings in this setting largely mirror those of the single-mode experiments. Under appropriate hyperparameter settings, Pipeline A-K consistently achieves superior Final Average Reward and Target Precision while maintaining competitive Overall Recall. However, the increased complexity of the dual-target task introduces a key distinction: higher variance across results. Crucially, this instability is disproportionately severe in Pipeline K-A, which exhibits drastic variance spikes. These empirical results confirm that the low-recall trap is exacerbated in multi-mode scenarios, underscoring Pipeline A-K as the significantly more robust and effective solution.

## C  DETAILS OF LLM EXPERIMENTAL SETUP

### C.1  IMPLEMENTATION DETAILS

All Large Language Model (LLM) experiments were implemented using PyTorch 2.6.0 (Paszke et al., 2019) and the HuggingFace Transformers library (Wolf et al., 2019). For efficient training of all models, we utilized DeepSpeed (Rasley et al., 2020) with bfloat16 precision. The alignment algorithms (PPO and DPO) were implemented using the TRL (Transformer Reinforcement Learning) library v0.9.6 (von Werra et al., 2020). All experiments were conducted over three independent seeds, and the results reported in the main paper are the average of these runs.

### C.2  KNOWLEDGE DISTILLATION

Our knowledge distillation (KD) pipeline follows the methodology of Cha & Cho (2025), which involves a two-step process: data generation from a teacher model, followed by student model training on the generated data.

**Data Generation.**  To create a training dataset for a student model, we generate text from a teacher model. We start with the simple prompt "The" and generate `num_samples = 10,000,000` sequences of up to `max_length=512` tokens. The generation process uses nucleus sampling with

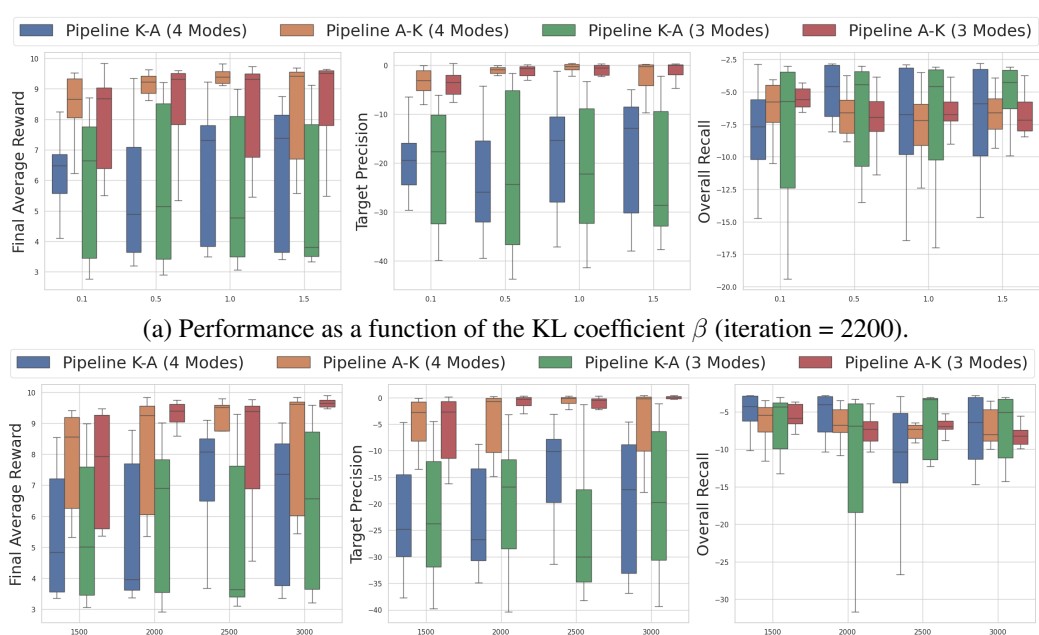

(a) Performance as a function of the KL coefficient $\beta$ (iteration = 2200).

(b) Performance as a function of the number of training iterations ($\beta = 1.0$).

Figure 10: RL (PPO) experiments with multiple target modes comparing Pipeline K-A (yielding $p''_{\text{KA}}$) and Pipeline A-K (yielding $p''_{\text{AK}}$). The boxplots summarize results over 20 seeds, sweeping across (a) the KL coefficient $\beta$ and (b) the number of training iterations.

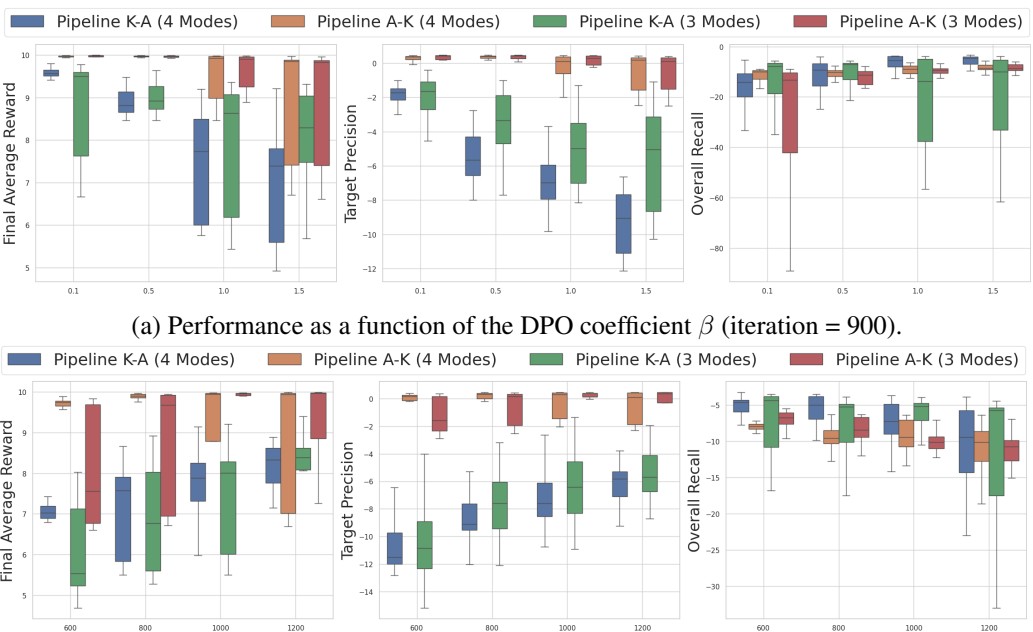

(a) Performance as a function of the DPO coefficient $\beta$ (iteration = 900).

(b) Performance as a function of the number of training iterations ($\beta = 1.0$).

Figure 11: On-policy DPO experiments with multiple target modes comparing Pipeline K-A (yielding $p''_{\text{KA}}$) and Pipeline A-K (yielding $p''_{\text{AK}}$). The boxplots summarize results over 20 seeds, sweeping across (a) the coefficient $\beta$ and (b) the number of training iterations.

top_p=1.0 and a specified temperature $\tau$. As described in the manuscript, we use different temperatures to create our various models: $\tau = 1.0$ for the dataset to train the high-recall $p'$ model, $\tau = 0.95$ for the low-recall $p''$ model, and a low temperature of $\tau = 0.8$ for the oracle $p_{\text{target}}$. In

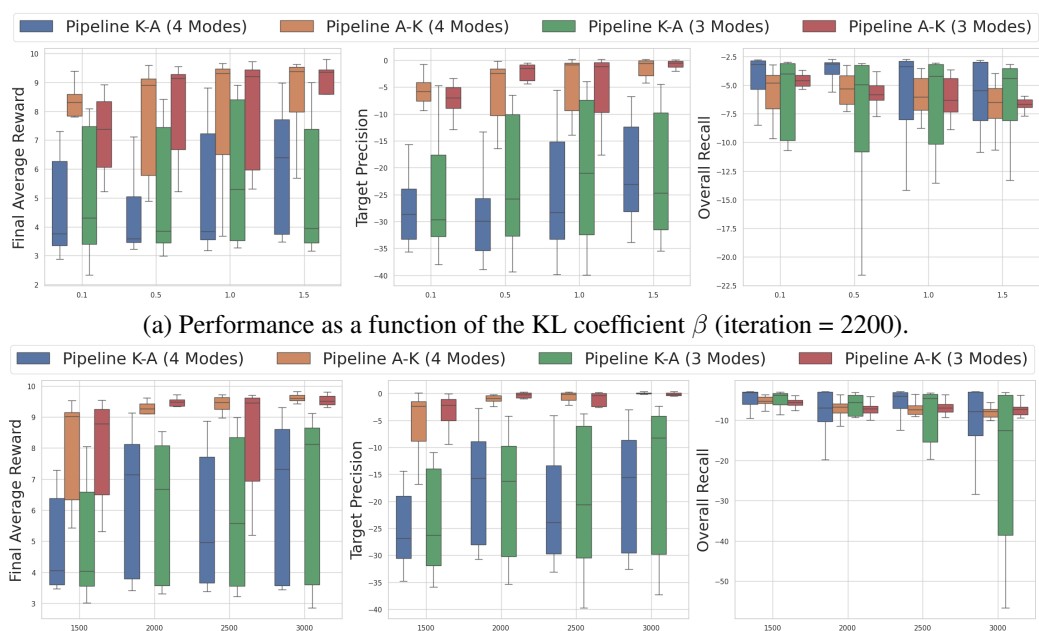

(a) Performance as a function of the KL coefficient $\beta$ (iteration = 2200).

(b) Performance as a function of the number of training iterations ($\beta = 1.0$).

Figure 12: RL (GRPO) experiments with multiple target modes comparing Pipeline K-A and Pipeline A-K. The boxplots summarize results over 20 seeds, sweeping across (a) the KL coefficient $\beta$ and (b) the number of training iterations. We report three key metrics: Final Average Reward, Target Precision, and Overall Recall.

a case of generating validation dataset, we sample `num_samples = 100,000` with $\tau = 1.0$ for each trained model by KD.

**Student Model Training.** The student model is trained on the dataset generated by its teacher using a standard causal language modeling objective. We use the AdamW optimizer with a learning rate of `5e-4` and betas of $(0.9, 0.95)$. The learning rate is managed by a custom Warmup-Stable-Decay (WSD) scheduler, with a warmup phase of 1% and a decay phase of 20% of the total training steps. The models are trained for a fixed number of epochs, with a global batch size of `mini_batch_size * world_size`, where `mini_batch_size` is 64.

## C.3 PREFERENCE ALIGNMENT (PPO & DPO)

We used the TRL library for our PPO and on-policy DPO implementations. All alignment experiments generate text from the prompt "`The`" with a generation temperature of $\tau = 1.0$ up to `max_length=128` tokens.

**PPO Implementation.** Our PPO setup uses TRL's `PPOTrainer` with an `AutoModelForCausalLMWithValueHead`, which combines the actor and critic into a single model. Key hyperparameters for our experiments include a learning rate of `1e-5`, a KL coefficient $\beta$ of `0.7`, a PPO batch size of `64`, and a mini-batch size of `8`. We train for `1` PPO epoch per batch.

**Reward Formulation for LLM Experiments.** Our primary objective was to design a reward function that is both simple and effective. Initially, we employed a direct reward based on the Negative Log-Likelihood (NLL) under the oracle target model, $p_{\text{target}}$. However, unconstrained maximization of this simple metric consistently led to mode collapse, where the policy converged to generating a narrow set of repetitive, high-probability sequences. We found that standard mitigation strategies, such as reward clipping, were ineffective; they either failed to prevent collapse or

hindered the model from maximizing the reward entirely. To address this structural instability, we implemented a two-stage "reward folding" mechanism. First, a base reward is calculated over the response tokens:

$$R_{\text{base}}(x, y) = 10.0 \cdot \exp(-C \cdot \text{NLL}(y|x; p_{\text{target}})), \tag{9}$$

where $C = 0.5$ is a scaling factor. Second, we apply the folding mechanism. Unlike clipping, this function penalizes a model for achieving an excessively high reward (*i.e.*, a reward above the threshold $\tau$, which signals mode collapse). By "reflecting" this reward to a lower value, it alleviates the policy from collapsing, successfully stabilizing the training.:

$$R_{\text{final}}(x, y) = \begin{cases} R_{\text{base}}(x, y) & \text{if } R_{\text{base}}(x, y) \leq \tau \\ 2\tau - R_{\text{base}}(x, y) & \text{if } R_{\text{base}}(x, y) > \tau \end{cases}. \tag{10}$$

This reflection mechanism actively discourages the policy from exploiting the reward oracle, successfully stabilizing the training process. The threshold $\tau = 8.4636$ was determined empirically in a preliminary study. We generated a large corpus of sentences from $p_{\text{target}}$ itself, computed their base reward distribution, and selected the 90th percentile as the threshold $\tau$.

**On-Policy DPO Implementation.** Our on-policy DPO implementation uses TRL's `DPOTrainer` in an online fashion. At each of the `2000` online iterations, the current policy generates a pool of `128` responses (`2 × batch_size`). These responses are then paired up and labeled to create `64` preference pairs for training. The trainer then performs `16` gradient updates on this newly generated batch of preferences. Key hyperparameters include a learning rate of `5e-6`, a KL coefficient $\beta$ of `0.7`, a mini-batch size of `4`, and `2` gradient accumulation steps. During tokenization for the DPO loss, the prompt portion of the labels is masked with `-100` to ensure the loss is calculated only on the response tokens.

**Preference Generation for DPO.** The preference labeling is performed by the oracle model $p_{\text{target}}$. For each pair of responses, we calculate their NLL under $p_{\text{target}}$. The response with the lower NLL (higher probability) is labeled as "chosen ($y_w$)", and the other is labeled as "rejected ($y_l$)".

## C.4 MODEL SELECTION FOR LLM ALIGNMENT

As noted in the manuscript, selecting a final model based solely on the maximum achievable reward can be misleading. During alignment, both PPO and on-policy DPO may over-optimize for the reward function, leading to a collapse in output diversity where the model repeatedly generates near-identical high-reward sentences (Gao et al., 2023; Kirk et al., 2024). This phenomenon, while optimal from a pure reward maximization perspective, is undesirable for practical applications. Therefore, we adopted a principled early stopping approach to select model checkpoints that demonstrate a strong alignment signal without sacrificing diversity. Our specific criteria for PPO and DPO are detailed below, with illustrative performance graphs in Figure 13 and Figure 14.

**PPO Model Selection.** For PPO, we monitored the mean and standard deviation of the rewards obtained by the policy at each evaluation step. As shown in Figure 13, the mean reward generally increases throughout training. Initially, the reward standard deviation rises in tandem with the mean reward, indicating that the policy is exploring diverse, high-reward strategies. However, after a certain point, the standard deviation declines sharply. Our analysis confirmed that this inflection point marks the onset of mode collapse, where the model begins to repeatedly generate the same few high-reward sentences. While the mean reward may continue to increase to its maximum value past this point, this is achieved at the cost of diversity. To balance the objectives of high reward and response diversity, we selected the model checkpoint from an iteration where the mean reward was high, and critically, before the sharp decline in reward standard deviation.

**DPO Model Selection.** For on-policy DPO, we tracked four key metrics over the training iterations, as depicted in Figure 14: the average rewards of accepted ($y_w$) and rejected ($y_l$) responses, the margin between them, and the classification accuracy on newly generated preference pairs. An ideal model should not only maximize the reward of chosen responses but also maintain a clear distinction between preferred and dispreferred outputs. As shown in the figure, the rewards for both accepted and rejected answers increase during training, but the accepted reward rises more steeply,

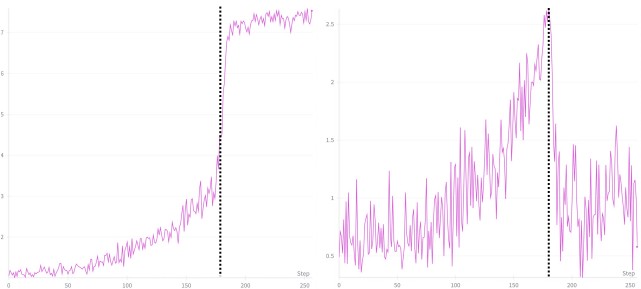

Figure 13: PPO Model Selection Trajectories. (a) Mean reward and (b) reward standard deviation over training iterations. The vertical dashed line indicates the selected checkpoint, which achieves a high mean reward while retaining high reward variance, thus avoiding mode collapse.

leading to a widening reward margin. However, after a certain point, this margin begins to decline sharply. This decline signals the onset of mode collapse, where the policy starts generating only a few, near-identical high-reward sentences. Consequently, as the generated chosen and rejected responses become nearly indistinguishable, the preference pairs become uninformative, causing the classification accuracy to collapse. To balance the objectives of high reward and response diversity, we selected the model checkpoint from the iteration that maximized the reward margin, capturing the point of peak preference discrimination before the onset of mode collapse.

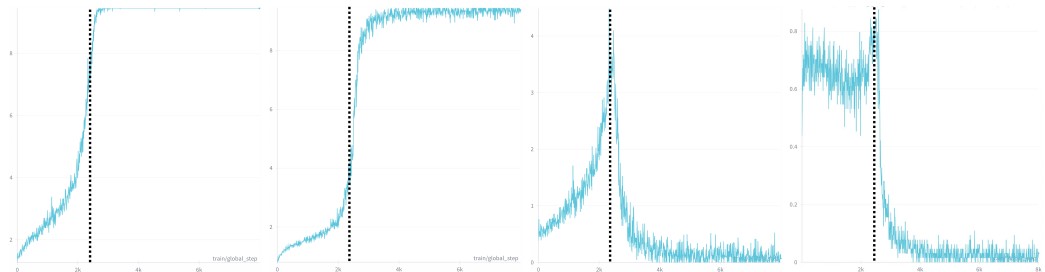

Figure 14: DPO Model Selection Trajectories. (a) Average reward of accepted answers, (b) average reward of rejected answers, (c) the margin between them, and (d) classification accuracy on preference pairs over training iterations. The vertical dashed line marks the selected model at the point of peak margin, indicating the point of peak preference discrimination before the onset of mode collapse.

## D ADDITIONAL EXPERIMENTAL RESULTS WITH LLMS

To demonstrate the tunability of the precision-recall trade-off within Pipeline A-K, we conducted additional experiments varying the distillation temperature during the final KD stage. The results are summarized in Figure 15. By comparing these results with the baseline presented in Figure 5 (where $\tau = 0.95$), we observe that adjusting $\tau$ effectively modulates the student model's generative behavior:

- Baseline ($\tau = 0.95$): In the main experiments, PPO achieved a Precision of -2.1573 and Recall of -4.2113, while DPO achieved a Precision of -1.2187 and Recall of -4.4132.

- Lowering Temperature (PPO, $\tau = 0.925$): Reducing the temperature resulted in a higher Precision of -1.9880 (compared to -2.1573) and a lower Recall of -4.2371 (compared to -4.2113). This confirms that lowering $\tau$ sharpens the distribution, improving precision at the cost of recall.

- Raising Temperature (DPO, $\tau = 0.975$): Increasing the temperature resulted in an improved Recall of -4.3346 (compared to -4.4132) and a lower Precision of -1.5623 (compared to -1.2187). This confirms that raising $\tau$ broadens the distributional coverage, improving recall at the cost of precision.

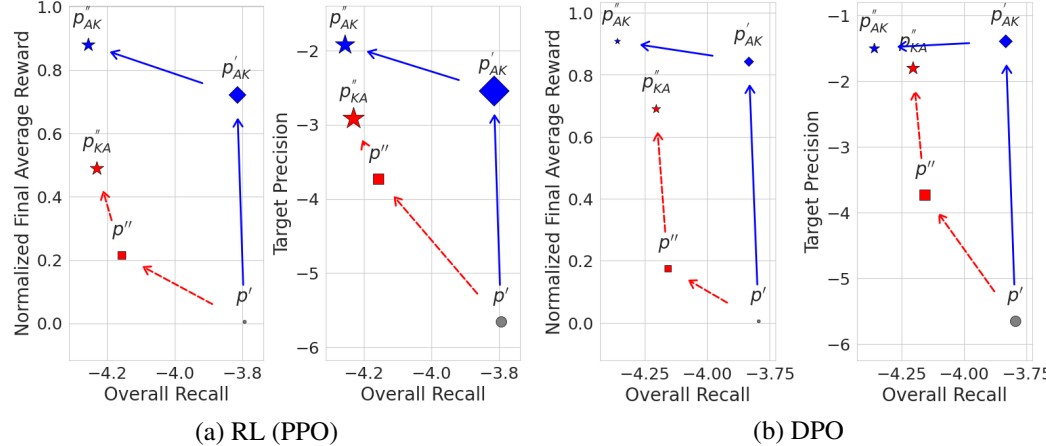

(a) RL (PPO)  (b) DPO

Figure 15: LLM alignment trajectories comparing Pipeline K-A (red) and Pipeline A-K (blue). The plots show the evolution of models for (a) RLHF (PPO) and (b) DPO in the performance space defined by Overall Recall (x-axis) and target-oriented metrics (y-axis). Each marker indicates the mean performance over three seeds, with its size proportional to the cross-seed standard deviation (instability). Arrows depict the pipeline evolution. In the process of KD for $p''_{AK}$, we set $\tau = 0.925$ for PPO and $\tau = 0.975$ for DPO.

These empirical findings validate our claim that Pipeline A-K offers a tunable trade-off. By selecting an appropriate $\tau$ during the final distillation step, practitioners can explicitly control the balance between sample quality (precision) and distributional coverage (recall) to suit their specific alignment goals.

# E   LIMITATIONS

Our work establishes the distributional recall of the reference model as a first-order design choice in preference alignment, demonstrating the structural failures of the common KD → Align pipeline. While our findings are validated through controlled synthetic experiments and realistic LLM setups, we acknowledge several limitations that open promising avenues for future research. The conclusions presented in this paper are subject to the following limitations:

- **Scope of Models and Tasks:** Our empirical validation primarily utilized the SmolLM2 family of models, which are relatively small by current standards. While this controlled setting was ideal for isolating the low-recall trap mechanism, these findings need to be validated on larger, state-of-the-art models (*e.g.*, 70B parameters) where distillation trade-offs and alignment dynamics may differ.

- **Synthetic Nature of the Alignment Target:** The alignment target in our LLM experiments was defined by a reward oracle derived from another distilled model ($p_{\text{target}}$). This provided a perfect, noise-free preference signal, which is rarely the case in real-world scenarios. Future work should investigate whether our conclusions hold when aligning with noisy and diverse human preferences, particularly in complex domains like creative writing or safety-critical applications where desirable behaviors are often rare and difficult to specify.

- **Simplicity of Prompts:** To maintain a controlled experimental environment, we used a simple, generic prompt ("The") for text generation. The dynamics of the low-recall trap might be exacerbated or altered when using a wider, more complex distribution of user-facing prompts.

# F   USE OF LARGE LANGUAGE MODELS

The authors utilized Large Language Models (LLMs) to assist in the revision process of this manuscript. Specifically, after the core scientific contributions, experimental results, and initial drafts were completed by the authors, LLMs were used for tasks such as proofreading, refining

sentence structure, and polishing the language for clarity and readability. The authors take full responsibility for all content in this paper.

## G  ETHICS STATEMENT

The authors acknowledge and adhere to the ICLR Code of Ethics. Our work focuses on the fundamental principles of preference alignment and knowledge distillation in machine learning models. The experiments are conducted on a synthetic Mixture-of-Gaussians dataset and with publicly available language models from the `SmolLM2` family. This research does not involve human subjects, private user data, or the release of new datasets. We believe the proposed pipeline for improving alignment efficiency does not introduce new or direct societal harms.

## H  REPRODUCIBILITY STATEMENT

We are committed to ensuring the reproducibility of our work. All experimental details, including model configurations, training hyperparameters, and dataset preparation for both the Mixture-of-Gaussians and LLM experiments, are thoroughly documented in the Appendix (Sections A and C). Our implementation relies on standard open-source libraries, including PyTorch, Hugging Face Transformers, and TRL. We also attached our implementation code as supplementary materials.

