# OpenReview forum: "Why Alignment Must Precede Distillation: A Minimal Working Explanation"
_ICLR.cc/2026/Conference — Submitted to ICLR 2026_

### Official Review · Reviewer_NvRw · 2025-10-27

**Soundness:** 3
**Presentation:** 3
**Contribution:** 3
**Rating:** 4
**Confidence:** 3

**Summary:**

This paper tackles the common practice of performing knowledge distillation (KD) before alignment. The paper provides evidence that alignment followed by KD is better. They show this via two sets of experiments. First, they use a mixture of gaussian example to show this. Second, they use a small LM example. They show that in general KD-then-alignment is worse than alignment-then-KD in terms of precision and variance. The former often shows a better recall, though, but this comes with a huge sacrifice in precision.

**Strengths:**

- The paper's main message is consistent.
- To me, the really important point is not that K-A is better than A-K, because surely KD looses the model capacity. I think the common sense is that one should do KD at the final step in the training pipeline. The point that matters is that K-A was standard (I did not know this before) and that the gap between the two is meaningfully large as demonstrated by this paper..

**Weaknesses:**

- It is a bit confusing that there is a recall trap but at the same time, often K-A achieves a better recall (with large beta). Could you please explain this?
- To me, the set of experiments performed is far from being comprehensive. For example, a different base LLM would be great.
- Also, I was not quite convinced that the reward setting (9) is reasonable? Why do we need that reflection? Something like clipping would make more sense to me.
- other minor comments
	- Eq 4: the target precision is defined to be the same as the overall precision. is this typo?
	- L424: p^* is redefined (previously defined in line 418). Please avoid this to improve readability.
	- L462-470: The authors imply that the disadvantage of $p_{AK}$ (having low overall recall) can be overcomed by adjusting distillation temperature. It would be great to see this confirmed empirically.

**Questions:**

- How is the reward (9) justified? Why apply the 'reflection' there?
- For off-policy DPO + K-A, one could say that the $\pi_{ref}$ can be set to be the SFT model before KD. I wonder if this can help avoid the issues that K-A is encountering.
- For K-A + PPO, one could use $\pi_{ref}$ in the regularizer as the SFT model before KD. I wonder if this can help avoid the issues that K-A is encountering.

---

> ### Author Response · Authors · 2025-11-17
>
> We thank the reviewer for their detailed comments. We have prepared a response to each concern below. A revised version of our paper reflecting these points will be uploaded as soon as possible during the discussion period, at which time we will provide a summary of the changes.
>
> ## Weaknesses
>
> ### 1. K-A achieves a better recall (with large beta)
>
> While our results do show cases where Pipeline K-A achieves higher recall, we argue this is "misleading recall." As discussed in Lines 370-374 of the manuscript, this phenomenon occurs when K-A's Target Precision collapses.
>
> Specifically, under a large $\beta$, the strong regularization causes the K-A pipeline to fail the alignment objective. Because the model is not successfully focused on the target mode (i.e., it has very low precision), its distribution remains more spread out, resulting in a higher, "misleading" recall score. Both the PPO and DPO results in Figure 3 and 4 clearly illustrate this: K-A's recall increases only as its precision is severely degrading.
>
> In contrast, Pipeline A-K successfully aligns (maintaining high target precision) and maintains a competitive recall, demonstrating its overall superiority.
>
> ### 2. To me, the set of experiments performed is far from being comprehensive.
>
> We would like to respectfully clarify what we believe is a misunderstanding of our experimental setup. Our goal is not to report performance on popular benchmarks.
>
> Instead, we designed a controlled setup to analyze the distributional changes (via precision and recall) among the GT, teacher/reference, and student/aligned models. This methodology allows us to obtain empirical answers to the procedural question of whether KD or Alignment should come last.
>
> Our conclusions are therefore not dependent on a specific large-scale model or a particular downstream task. They are derived from a more general, mechanism-level analysis of the LLM post-training process. In this regard, **we respectfully refer the reviewer to [Common response 1 for Reviewer RXEC and v1RN: Experimental setup of LLM experiments] for a detailed explanation of this experimental setting.** Also, note that this experimental setup is positively evaluated by the reviewer aqcD (Strength 2).
>
> The use of SmolLM2 1.7B, 360M, and 135M models was purposeful for this setup, allowing them to serve as the GT (p∗), teacher/reference model p′, and student/aligned model p′′, respectively. We believe our findings are general and not an artifact of model size. **We would like to emphasize that a similar setup with larger models would exhibit the same fundamental trends**.
>
> ### 3. I was not quite convinced that the reward setting (9) is reasonable? Why do we need that reflection?
>
> Our objective was to design the simplest, clearest reward function. For the LLM experiments, we initially used a basic reward based on the Negative Log-Likelihood (NLL) under the target model.
>
> However, as noted in Appendix C.3, this simple reward consistently led to "mode collapse," where the policy would output a few identical, high-reward sequences. We found that standard solutions, like reward clipping, were ineffective; they either failed to solve mode collapse or prevented the model from maximizing the reward at all.
>
> We implemented the "reward folding" mechanism (Eq. 9) specifically to solve this. Unlike clipping, this function penalizes a model for achieving an excessively high reward (i.e., a reward above the threshold $\tau$, which signals mode collapse). By "reflecting" this reward to a lower value, it alleviates the policy from collapsing, successfully stabilizing the training.
>
> ### 4.1 - Eq 4: the target precision is defined to be the same as the overall precision. is this typo?
>
> This is not a typo, but the notation is subtle. Overall Precision uses p∗ (the 8-mode ground-truth distribution). Target Precision uses p⋆ (the single-mode target distribution). They measure precision against the GT and the desired target, respectively.
>
> ### 4.2 - L424: p∗ is redefined (previously defined in line 418). Please avoid this to improve readability.
>
> This is the same point as above. We acknowledge that the notation (p∗ vs. p⋆) is confusing. We will revise the notation in the final paper to make the distinction between the Ground-Truth distribution and the Target-Mode distribution much clearer.
>
> ### 4.3 - L462-470: The authors imply that the disadvantage of (having low overall recall) can be overcome by adjusting distillation temperature. It would be great to see this confirmed empirically.
>
> Thank you for your suggestion. We are currently running these experiments and will add the results (showing the recall/precision trade-off via temperature in the A-K pipeline) to the supplementary materials in the revised paper. We will post an update when it is available.

---

> ### Author Response · Authors · 2025-11-17
>
> ## Questions
>
> ### Q.1 How is the reward (9) justified? Why apply the 'reflection' there?
>
> Please see our response to Weakness 3. The reflection was a necessary mechanism to prevent mode collapse during LLM alignment training.
>
> ### Q.2 For off-policy DPO + K-A, one could say that the π_ref can be set to be the SFT model before KD...
> ### Q.3 For K-A + PPO, one could use π_ref in the regularizer as the SFT model before KD...
>
> These two questions propose the same, insightful "hybrid" scenario:
>
> 1. Pipeline: K-A (i.e., pi_\theta is the small, KD model).
> 2. Reference: pi_ref is the large, high-recall model (before KD).
> 3. Question: Does this avoid the low-recall trap?
>
> This is a interesting approach. While this approach might mathematically resolve the "exploding penalty" term, we argue it would fail by creating a more fundamental "anchor mismatch" problem.
>
> In PPO and DPO, the pi_ref term is not just an arbitrary offset; it serves as the anchor to regularize the policy pi_\theta against its own initial state (pi_ref=pi_θ_init).
>
>
> The proposed hybrid approach completely breaks this identity. pi_\theta(the small model) and pi_ref(the large model) are now two entirely different models. For example in DPO, this forces the objective function to pursue two conflicting goals simultaneously:
>
> 1. [DPO Goal]: Learn the preference (R(y_w)>R(y_l)).
> 2. [KD Goal]: Stay close to the distribution of pi_ref (the large model).
>
> This setup is no longer stable alignment; it has become an unstable, simultaneous "preference learning + cross-architecture distillation" algorithm. This is not what PPO/DPO were designed for and would likely fail to optimize.
>
> ----
>
> We thank the reviewer for this highly constructive feedback. We have provided our responses and are incorporating these clarifications into our revised manuscript. We will upload the new version as soon as it is complete and are happy to address any further questions.

---

> ### Author Response · Authors · 2025-11-21
> **Author response after revision**
>
> We sincerely thank you for your valuable feedback. Based on your comments, we have implemented the following revisions to clarify the main contribution of our paper and strengthen our empirical evidence.
>
> * **Justification for the Reward Setting (Eq. 9):**
>     Building on our initial response, we have expanded the description of the reward function in the Appendix C.4. We added a detailed explanation justifying the use of the reflection mechanism, clarifying that it serves to penalize over-optimization, rather than simply clipping the reward.
>
> * **Impact of Distillation Temperature on $p_{\text{AK}}$:**
>     We have added new experimental results in Figure 15 of Appendix D to demonstrate how the distillation temperature ($\tau$) modulates the performance of the $p_{\text{AK}}$ pipeline.
>     By comparing these new results with the baseline in Figure 5 (using $\tau=0.95$) of the manuscript, we confirm that adjusting $\tau$ during the final KD stage effectively controls the precision-recall trade-off:
>     * Baseline (Figure 5, $\tau=0.95$): PPO achieved Precision: -2.1573 / Recall: -4.2113. DPO achieved Precision: -1.2187 / Recall: -4.4132.
>     * Temperature Modulation Experiments:
>         * PPO with lower $\tau=0.925$: Resulted in **higher Precision (-1.9880)** and lower Recall (-4.2371). This confirms that lowering the temperature sharpens the distribution, improving precision at the cost of recall.
>         * DPO with higher $\tau=0.975$: Resulted in lower Precision (-1.5623) and **improved Recall (-4.3346)**. This confirms that raising the temperature broadens coverage, improving recall at the cost of precision.
>     * These findings empirically validate our claim that Pipeline A-K offers a tunable trade-off. We plan to integrate these results into the manuscript of the camera-ready version to explicitly demonstrate this controllability.
>
> * **Minor Comments (Notation):**
>     To eliminate potential confusion with the ground truth distribution, we have globally revised the notation $p^\star$ to **$p_{\text{target}}$**.
>
> We hope that these revisions and additional experiments satisfactorily address your concerns. We are committed to further refining the manuscript for the camera-ready version to ensure that our core contributions are presented with maximum clarity. Thank you again for your insightful comments.

---

### Official Review · Reviewer_v1RN · 2025-10-29

**Soundness:** 3
**Presentation:** 3
**Contribution:** 2
**Rating:** 4
**Confidence:** 3

**Summary:**

This paper investigates a critical design choice in the pipeline for creating efficient, aligned language models. The authors identify a potential flaw in the common practice of first distilling a large model into a smaller one and then performing preference alignment (`KD → Align`). They term this flaw the "low-recall trap," arguing that distillation prunes rare but desirable behaviors, making them difficult to recover during alignment. The paper proposes reversing the pipeline to `Align → KD`, where a large, high-recall model is first aligned and then distilled. Through experiments on a Mixture-of-Gaussians toy problem and the SmolLM2 model family, the authors demonstrate that their proposed pipeline yields models with superior performance on target metrics, higher rewards, and greater training stability.

**Strengths:**

The paper possesses several key strengths. It addresses a problem in the practical deployment of LLMs, challenging a common practice with a intuitive concept: the "low-recall trap." The motivation is clear, and the proposed Align → KD alternative is simple. Furthermore, the empirical validation strategy, which uses both a controlled synthetic experiment to isolate the mechanism and a practical LLM experiment to demonstrate its relevance, is methodologically sound and provides consistent evidence for the authors' claims within the tested scope.

**Weaknesses:**

While the paper addresses an important and interesting question in the practical application of LLM alignment, and its central thesis is compelling, I have several major concerns that, in my view, should be addressed before the paper can be considered for publication. The core idea is valuable, but the work in its current form would need to substantially strengthen its evidence and analysis regarding the following points: **(1)** the generalizability of the findings beyond a highly constrained and simplified experimental setting; **(2)** the reliance on idealized assumptions about the alignment signal that differ from noisy real-world scenarios; **(3)** the omission of the critical aspect of computational cost, which is the primary motivation for the very practice it critiques; and **(4)** the lack of discussion of crucial prerequisites and potential alternative solutions.

**Major Weaknesses**

**The Generalizability of the Findings**

A primary concern is that the experimental validation, while well-executed within its scope, is conducted in a setting that may not be representative of modern, large-scale LLM applications. For the paper's claims to be broadly impactful, the following points need to be addressed:

*   **Model Scale:** The experiments are limited to the SmolLM2 family (up to 1.7B parameters). It is difficult to assume that conclusions drawn from these small-scale models will hold for the state-of-the-art models (e.g., 30B+ parameters) where alignment and distillation challenges are most pronounced. The paper would be much stronger if it demonstrated that this trend persists on at least one larger, more capable model.
*   **Task Complexity:** The LLM alignment task is based on text generation from a single, generic prompt ("The") with an "oracle reward model". This oversimplification is a significant departure from real-world alignment, which involves complex instruction-following and reasoning. To make a convincing case, the authors should consider evaluating their proposed pipeline on established and more complex alignment benchmarks, such as **MT-Bench, AlpacaEval, or Arena-Hard**.

**Reliance on Idealized Assumptions**

Real-world alignment is fundamentally shaped by noisy and inconsistent human feedback. This raises a critical, unaddressed question:

*   **Robustness to Noisy Preferences:** Does the performance advantage of the `Align → KD` pipeline hold when the reward signal is weak, stochastic, or noisy, as is typical with human data? An investigation into this would be crucial, as the high cost of aligning a large model may not be justified if the benefits diminish significantly under realistic conditions.

**Omission of Computational Cost Analysis**

A crucial aspect that the paper does not sufficiently address is the computational cost, which is the primary driver behind the `KD → Align` paradigm in the first place. While the paper convincingly argues for the performance benefits of `Align → KD`, its practical viability is left unevaluated.

*   The paper would be significantly strengthened by including a cost-benefit analysis. A discussion of the trade-offs in terms of computation resource metrics is essential for practitioners to gauge whether the performance gains of `Align → KD` justify its substantially higher upfront computational cost.

**Insufficient Exploration of Prerequisites and Alternatives**

Finally, the analysis would be more robust if it considered the boundary conditions of its claims and explored potential alternatives more deeply.

*   **The Role of SFT Quality:** The proposed pipeline's success is critically dependent on starting with a high-recall SFT model. The paper would benefit from a discussion of how the quality and diversity of the SFT data impact the initial recall and, consequently, the effectiveness of the entire `Align → KD` pipeline.
*   **Mitigation Strategies for the "Low-Recall Trap":** The paper presents the `Align → KD` pipeline as the solution to the "low-recall trap" but does not explore whether the trap can be mitigated within the standard `KD → Align` framework. For instance, could a more sophisticated data-sourcing strategy for distillation that improves the diversity and coverage of the training data alleviate the problem? Comparing against such an enhanced baseline would provide a more complete picture.

I am willing to raise my score if the authors can satisfactorily address the weaknesses I have raised.

**Questions:**

See weaknesses

---

> ### Author Response · Authors · 2025-11-17
>
> We thank the reviewer for their detailed comments. We have prepared responses to each concern below. A revised version of our paper reflecting these points will be uploaded as soon as possible during the discussion period, at which time we will provide a summary of the changes.
>
> ## Weaknesses
>
> ### 1. The Generalizability of the Findings.
>
> We would first like to respectfully clarify what we believe is a fundamental misunderstanding of our experimental design. Our experimental setup is not intended to report benchmark scores on large-scale LLMs.
>
> Instead, we have designed a controlled setup to analyze the distributions of the ground-truth (GT;p*), teacher/refernce model (p'), and student/aligned model (p'') models via precision and recall. This methodology allows us to obtain empirical answers to the question of pipeline ordering (KD vs. Align) by observing how these distributional properties change.
>
>
> Our conclusions are therefore not dependent on a specific large-scale model or a particular downstream benchmark. They are derived from a more general analysis of the post-training process itself from a precision-recall perspective. In this regard, **we respectfully refer the reviewer to [Common response 1 for Reviewer RXEC and v1RN: Experimental setup of LLM experiments] for a detailed explanation of this experimental setting.**
>
> Given this experimental framework, we hope the reviewer can understand why large-scale experiments on complex reasoning tasks (MT-bench, AlpacaEval, etc.) are not directly applicable. Our contribution lies in the mechanism we've isolated, and we kindly ask the reviewer to reconsider the distinct nature and contribution of our setup.
>
>
> ### 2. Reliance on idealized assumptions
>
> We want to emphasize that our paper demonstrates that even under the ideal conditions of a perfect oracle reward function, the KD $\rightarrow$ Align pipeline is structurally flawed and inferior to the Align $\rightarrow$ KD pipeline.
>
> Based on this conclusion, we can naturally project that this core tendency will hold in a "noisy" real-world setting. As long as the noisy human feedback is an unbiased, best-effort approximation of a true reward function, the underlying "low-recall trap" mechanism, which punishes the recovery of rare modes, will still be present in the KD $\rightarrow$ Align pipeline. The Align $\rightarrow$ KD pipeline, by avoiding this trap mechanistically, should remain the superior approach.
>
>
> ### 3. Omission of computational cost analysis
>
> We must first emphasize that our paper's primary contribution is not to propose a specific algorithm and demonstrate its cost-benefit. Instead, we provide an empirical answer to a general procedural question: in which order should KD and Alignment be performed to obtain high-quality small models?
>
> In our experiments, both KD $\rightarrow$ Align and Align $\rightarrow$ KD use the exact same KD methods and Alignment algorithms (DPO, PPO). Therefore, a direct comparison of algorithmic costs is not the central issue.
>
> It is self-evident that our proposed Align $\rightarrow$ KD pipeline incurs a higher alignment cost because the alignment step is performed on a larger model. We explicitly acknowledge this. However, this "cost-first" thinking is precisely the "common practice" we are challenging (Refer to Lines 54-59 of the manuscript). Our results demonstrate that while KD $\rightarrow$ Align may appear more cost-effective, it is a "false economy" that leads to a suboptimal, structurally flawed outcome (i.e., low precision and reward). Conversely, Align $\rightarrow$ KD, while more expensive at the alignment stage, is the path that successfully produces the high-quality student model (i.e., high precision and reward; See Figs 3,4, and 5 of the manuscript). We believe this conclusion is clear without a line-item analysis of the cost difference.

---

> ### Author Response · Authors · 2025-11-17
>
> ### 4. Insufficient exploration of prerequisites and alternatives
>
> We believe this weakness also stems from a misunderstanding of our experimental design. **We again kindly refer the reviewer to [Common response 1 for Reviewer RXEC and v1RN: Experimental setup of LLM experiments] for a full explanation of our controlled setup.**
>
> > The Role of SFT Quality: ...The paper would benefit from a discussion of how the quality and diversity of the SFT data impact the initial recall...
>
> Our work uses a controlled experimental setup (e.g., the simple prompt "The") precisely to avoid confounding variables like SFT data diversity. This controlled setup is a strength, as it allows us to isolate the effect of the pipeline order itself. Note that this is evaluated as a strength of this paper by Reviewer aqcD (Strength 2).
>
> By using our experimental setup, we can analyze the model's capability directly through the lens of distributional recall. As a result, we reported that Align $\rightarrow$ KD is a generally more robust protocol, with findings that are not contingent on the specific properties of the dataset.
>
> > Mitigation Strategies for the "Low-Recall Trap": The paper presents the Align → KD pipeline as the solution ... but does not explore whether the trap can be mitigated within the standard KD → Align framework.
>
> This is a key point. Our results, supported by [*Why Knowledge Distillation Works in Generative Models: A Minimal Working Explanation, Cha & Cho, NeurIPS 2025*], show that once KD is performed, a loss of recall is inevitable. This is a fundamental precision-recall trade-off inherent to compression.
>
>
> While more sophisticated KD methods or temperature tuning might increase the recall of the student model, this will necessarily come at the cost of precision due to the student's limited capacity [*Why Knowledge Distillation Works in Generative Models: A Minimal Working Explanation, Cha & Cho, NeurIPS 2025*]. If the goal is a high-quality (i.e., high-precision and competitive-recall) student model, our findings confirm it is far more effective to use the Align $\rightarrow$ KD pipeline, even with a standard KD algorithm, than to try to invent a sophisticated KD method to save the flawed KD $\rightarrow$ Align process.
>
> ----
>
> We thank the reviewer again for their constructive comments, many of which focused on real-world scenarios. We have centered our responses on clarifying the intent and value of our specific experimental setup. We hope these clarifications resolve the reviewer's concerns and highlight the contribution of our work. The revised manuscript is in progress and will incorporate these clarifications to minimize ambiguity. We will post an update upon its completion and are available for any further questions.

---

> ### Comment · Reviewer_v1RN · 2025-11-28
>
> I appreciate the author's detailed response and will raise my score.

---

### Official Review · Reviewer_XYEC · 2025-10-30

**Soundness:** 2
**Presentation:** 2
**Contribution:** 3
**Rating:** 2
**Confidence:** 3

**Summary:**

This paper challenges the common practice of performing preference alignment after knowledge distillation (KD) in model training. The authors argue that distilling before alignment reduces the reference model's distributional recall, limiting its ability to learn and align rare but desirable behaviors. They propose reversing the standard pipeline, i.e., aligning first, then distilling (Align -> KD),  to preserve these behaviors. The paper contributes:
1. A theoretical explanation of how the reference model's recall constrains alignment objectives.
2. Empirical validation using a controlled Mixture-of-Gaussians setup and on the SmolLM2 language model family.
Overall, the paper highlights *alignment must precede distillation* to maintain alignment quality and model robustness.

**Strengths:**

This paper addresses a compelling and timely question: *should alignment precede or follow distillation for small models?* The authors provide clear empirical evidence showing that performing alignment before distillation leads to better outcomes. A key strength of the work lies in its thorough empirical validation, which includes both simulated experiments and on real language models, demonstrating the robustness and generalizability of the findings.

**Weaknesses:**

* I find the overall narrative of the paper somewhat inconsistent. The work appears to conflate two related but distinct questions:
     1. What is a good reference model for alignment?
     2. Which process should be performed first: alignment or distillation?

    Most of the theoretical discussion and motivation center on the first question, emphasizing that the base model should maintain high recall. However, I feel like this point is already fairly well recognized in the community. Although it's probably due to a different reason: as RL often leads to entropy or mode collapse, a base model with high output diversity is essential.

    In contrast, the second question is more about: which process hurts diversity more, alignment or distillation? Both alignment (RL) and distillation can significantly reduce the diversity of language model outputs. Therefore, when developing compact student models, it would be valuable to investigate which step should precede the other to best preserve quality and diversity. I believe the paper would be stronger and more engaging if it focused more directly on this question.
* For the MoG experiments, it would be more interesting if the target distribution consisted of two modes of $p^*$, as mode collapse is a persistent and unresolved challenge in alignment research.
* For the LLM experiments, I'm not sure why it's necessary to include language models of three different sizes. In practice, there's typically a single large, high-capability model, and the smaller ones are usually distilled from it.
* Minor points: The authors should revise the content that was directly copied and pasted from LLM outputs, especially the hyphens. For example, "Kullback-Leibler" always has a longer-than-usual hyphen. This is also the case for "precision-recall" on line 258 on page 5. Besides, an em dash as a long punctuation mark should be "---" in latex.

**Questions:**

1. I'm not sure whether the actual pipeline for obtaining smaller models follows the sequence "pretrain–SFT–KD–alignment." One key confusion is the blurred boundary between SFT and KD: both stages share the same training objective and use off-policy data. In practice, people don't seem to make a clear distinction between the two. If that's the case, performing RL or DPO before this stage seems somewhat unusual to me. Although the base model may exhibit high diversity, the probabilities of producing "good" responses could still be extremely small. In this case, issues like sampling and learning traps persist. Am I misunderstanding something?
2. The second question relates to the first one. After reading the entire paper, I'm still unclear about how to obtain high-quality small models. I guess this mainly stems from the confusion about difference between SFT and KD. It would be helpful if the authors could describe the pipeline more explicitly somewhere in the paper.

---

> ### Author Response · Authors · 2025-11-17
>
> We thank you for your detailed review and comments. We have prepared a response to each of your concerns below. A revised version of our paper reflecting these points will be uploaded during the discussion period, and we will notify you with an additional response at that time.
>
> ## Weaknesses
>
> ### 1. On the inconsistent narrative
>
> We thank the reviewer for this insightful observation. The reviewer is correct that our work addresses these two related questions. We agree that the premise "a high-recall reference model is preferable for alignment" (Question 1) is a recognized intuition, often motivated by the need to mitigate the mode collapse associated with RL. However, we would like to emphasize that our primary contribution is not to simply restate this intuition, but rather:
> 1. To identify a common, conflicting practice: performing KD $\rightarrow$ Align for efficiency, which uses a low-recall distilled model as the reference pi_ref (See Lines 180-185 of the manuscript).
> 2. To formalize the mechanism of its failure, we demonstrate that this specific practice creates a structural "low-recall trap" (See Section 3.4 of the manuscript). As our analysis shows, this is not just a general loss of diversity, but a specific mathematical barrier (e.g., exploding KL penalties in PPO or gradient starvation in DPO) that actively prevents the recovery of rare, desirable modes pruned during the initial KD step.
>
>
> This analysis directly provides the answer to the reviewer's "more engaging" second question. **Our empirical results (Figs. 3, 4, 5 of the manuscript) confirm that the Align $\rightarrow$ KD pipeline is robustly superior to KD $\rightarrow$ Align in terms of achieving high precision (and reward) and competitive recall**. This demonstrates that **it is far more effective to first align the high-recall model and then apply KD as the final step**, rather than attempting to align a distilled model that has already incurred this structural trap.
>
> ### 2. The MoG Experiment Suggestion.
>
> > it would be more interesting if the target distribution consisted of two modes of p*, as mode collapse is a persistent and unresolved challenge in alignment research.
>
> We agree that studying mode collapse in this setting is a valuable research direction. The specific goal of our current MoG experiment is to create a "minimal working explanation" to isolate the recoverability of a single, rare mode that was pruned during distillation. Our setup was designed to test the "low-recall trap" in its simplest form using a single target mode.
>
> However, following the reviewer's suggestion, we are conducting new experiments with a multi-mode target. We will include these new results in the revised version of our paper (likely in the Appendix) to further strengthen our findings.
>
>
> ### 3. LLM Experiment
>
> > I'm not sure why it's necessary to include language models of three different sizes. In practice, there's typically a single large, high-capability model, and the smaller ones are usually distilled from it.
>
> We believe a detailed answer here will help resolve the reviewer's related concerns in Weakness 1 and Questions 1 & 2 as well.
>
> In **[Common response 1 for Reviewer RXEC and v1RN: Experimental setting], we introduced detailed explanation on our experimental setting**. We hope it would be helpful to understand our experimental setting more clearly.
>
> ### 4. Issues in writing
>
> Thank you for pointing this out. We acknowledge the formatting issues (e.g., hyphens, em dashes) that occurs during polishing and will correct all of them in the revised paper.

---

> ### Author Response · Authors · 2025-11-17
>
> ## Questions
>
> ### 1. The actual pipeline for obtaining smaller models.
>
> > I'm not sure whether the actual pipeline... follows "pretrain–SFT–KD–alignment." One key confusion is the blurred boundary between SFT and KD... If that's the case, performing RL or DPO before this stage seems somewhat unusual...
>
> We thank the reviewer for this critical question, as it highlights a fundamental confusion that we must resolve. The reviewer's assessment stems from an understandable fusion of SFT (Supervised Fine-Tuning) and KD (Knowledge Distillation). We must clarify our terms:
>
> 1. SFT: We define SFT as fine-tuning a pretrained model on a curated dataset of high-quality, human-generated instructions/responses. Its goal is to teach the capability of instruction-following. The (**large**) model after this stage is the standard pi_ref for alignment  *[Training language models to follow instructions with human feedback, L. Ouyang et al., NeurIPS 2022]*.
> 2. KD: We define KD as the process of compressing a large teacher model (e.g., the pretrained or the SFT model) into a **smaller** student model.
>
> With these definitions, the two pipelines we compare are:
>
> 1.  **Pipeline K-A (De Facto Practice):** Pretrain ($\rightarrow$ SFT) $\rightarrow$ **KD** $\rightarrow$ **Align**
>     First, compress the **large** pretrained or SFT model into a **small** model. Then, use that small, low-recall model as its own pi_ref for RLHF/DPO alignment.
> 2.  **Pipeline A-K (Our Proposal):** Pretrain ($\rightarrow$ SFT) $\rightarrow$ **Align** $\rightarrow$ **KD**
>     First, align the **large**, high-recall pretrained or SFT model. Then, use that aligned, high-quality large model as a teacher to *distill* its capabilities into a **small** student [*Why Knowledge Distillation Works in Generative Models: A Minimal Working Explanation, Cha & Cho, NeurIPS 2025*].
>
> Given that our ultimate goal is to produce compact, purpose-specific LLMs, our experiments intentionally isolate the ordering of **KD** (for compactness) and **Alignment** (for specialization), omitting SFT for the direct comparison. We focus on this interaction from a precision-recall perspective for two primary reasons:
>
> 1.  These two processes represent the most fundamental operations: **KD** is the standard method for achieving model compactness, while **Alignment** (RLHF/DPO) is the standard method for steering a model toward a specific objective.
> 2.  In practice, many available "pretrained" models are themselves the product of distillation from a much larger model. It is a common workflow to apply SFT and/or Alignment directly to these already-compressed models, mirroring our K-A pipeline (refer to Lines 54-59 and 180-185 of the manuscript).
>
> Consequently, our paper's core contribution is the empirical validation of which *order* of these two stages, applying KD (for compactness) and alignment (for specialization), is superior for producing a high-quality small model.
>
> As the reviewer noted, some researchers (i.e., especially, researchers in the frontier labs) may hold an *empirical intuition* about this (e.g., "the base model should maintain high recall"). However, **our work provides the first concrete, empirical evidence and mechanism-level analysis (i.e., the "low-recall trap") that *validates* and *explains* this intuition.** We believe this formalization and empirical grounding is a valuable contribution, as acknowledged by Reviewer aqcD (Strength 1).

---

> ### Author Response · Authors · 2025-11-17
>
> ### 2.  I'm still unclear about how to obtain high-quality small models.
>
> Following our clarification in the Question above, our paper's direct answer to this question is to use the **Align $\rightarrow$ KD** pipeline.
>
> Our MoG and LLM experiments demonstrate that in the multi-stage process of producing a compact, aligned model (involving SFT, Align, and KD), the *order* of the alignment and distillation steps is critical. We show that placing **Alignment before Distillation** is the superior sequence for producing high-quality small models.
>
> Our experimental evidence (from both MoG and LLM validations) provides a two-part justification for this:
>
> 1. **KD $\rightarrow$ Align Fails**: When distillation is performed first, the resulting compact model suffers from the **"low-recall trap"**. This makes it difficult or, in some cases, mathematically prohibitive for the subsequent alignment stage to recover the desired high-reward behaviors (See Pipeline K-A in Figs 3, 4, and 5 of the manuscript).
> 2. **Align $\rightarrow$ KD Succeeds**: When alignment is performed first, it not only avoids this "low-recall trap" but also introduces a key practical benefit (See Pipeline A-K in Figs 3, 4, and 5 of the manuscript). The final distillation step becomes a controllable process where practitioners can (e.g., via temperature) tune the trade-off between precision and recall to produce a high-quality small model tailored to their specific needs (See Lines 494-502 of the manuscript).
>
> Based on these findings, our practical recommendation for real-world LLM development is:
>
> 1.  To avoid the "low-recall trap," alignment using RLHF or DPO should be performed on the highest-recall model available (i.e., the largest model, *before* any significant compression).
> 2.  Knowledge Distillation (KD) should be applied as the **final step** to compress this already-aligned, high-quality large model into the final compact student model.
>
> ----
>
> We hope this detailed clarification helps resolve the reviewer's concerns and the potential misunderstanding about our proposed pipeline. We are currently incorporating these clarifications into the manuscript and will upload a revised version as soon as it is complete. We thank the reviewer again for their constructive engagement and are happy to address any remaining questions.

---

> ### Author Response · Authors · 2025-11-21
> **Author response after revision**
>
> We sincerely thank you for your constructive feedback. In addition to the clarifications provided in our initial author response, we have implemented the following revisions in the manuscript to explicitly address your concerns regarding the paper's narrative and experimental scope.
>
> * **Improving the Overall Narrative:**
>     To clarify the logical flow and directly address your second question (regarding the optimal ordering of alignment and distillation), we made the following structural updates:
>     * **Section 3.4:** We introduced a new paragraph titled "**Premise: Distillation Reduces Distributional Recall**." This section explicitly establishes that performing KD first inherently yields a low-recall reference model. This addition provides a solid logical foundation, allowing the subsequent explanations of the **Sampling Trap** and **Learning Trap** to follow naturally.
>     * **MoG, LLMs Experiment Conclusions & Concluding Remarks:** We revised the conclusion of the MoG experiments (Lines 421–426) and of the LLM experiments (Lines 499-501), and the final **Concluding Remarks** to clearly show our answer to your second question: preserving recall (by aligning first) is crucial for stability and precision.
>
> * **Additional MoG Experiments (Two-Mode Setting):**
>     As requested, we have added experimental results for PPO (Figure 10), DPO (Figure 11), and GRPO (Figure 12) on a two-mode target setting in Appendix B.4.
>     * Setup: We designated modes 6 and 7 (both the original mode (7) and its immediate left neighbor (6) in Figure 1(c) of the manuscript) as the alignment targets. The oracle reward function was modified to measure the Euclidean distance between a sample and the *nearest* of the two target modes.
>     * Results: The findings in this multi-mode setting largely mirror those of the single-mode experiments. Under appropriate hyperparameter settings, Pipeline A-K consistently achieves superior Final Average Reward and Target Precision while maintaining competitive Overall Recall. However, the increased complexity of the dual-target task introduces a key distinction: higher variance across results. Crucially, this instability is significantly severe in Pipeline K-A, which exhibits drastic variance spikes. These results empirically confirm that the low-recall trap is exacerbated in multi-mode scenarios, establishing Pipeline A-K as the more robust and effective solution.
>     * We will include a more detailed discussion of these results in the final camera-ready version to further strengthen our empirical validation.
>
> * **Minor Points (Hyphens):**
>     We have carefully reviewed the manuscript and corrected the misuse of hyphens and dashes throughout the text, as suggested.
>
> We hope that these revisions, combined with our initial response, satisfactorily address the weaknesses and questions you raised. We remain committed to further polishing the manuscript for the camera-ready version to ensure our core contributions are communicated with maximum clarity. Thank you again for your valuable insights.

---

### Official Review · Reviewer_aqcD · 2025-10-31

**Soundness:** 3
**Presentation:** 1
**Contribution:** 3
**Rating:** 4
**Confidence:** 4

**Summary:**

The paper questions a common practice in LLM post-training. Given a pretrained model, practitioners often distill a large model into a smaller one and then perform preference alignment. This workflow is motivated by efficiency, since smaller models are cheaper and easier to fine-tune. However, the authors argue that this practice degrades performance in terms of distributional recall.
They claim that distillation tends to preserve only high-probability modes while pruning low-probability ones. As a result, the alignment that follows cannot recover desirable but rare behaviors if they fall outside the support of the distilled model. To address this issue, the authors propose reversing the order of operations: first align a (high-recall) pretrained model and *then* distill it, while acknowledging that this approach increases computational cost.
The authors support their argument with controlled experiments using a Mixture-of-Gaussians setup and additional experiments on large language models from the SmolLM2 family.

**Strengths:**

- The paper raises an important question in LLM post-training. The main message is intuitive and clearly communicated. Although the proposed solution is conceptually straightforward, the authors convincingly explain why it matters and why the opposite pipeline remains dominant in practice. The work is also valuable in how it formalizes the informal intuition that alignment depends on the reference model’s coverage. The analysis of how "low-recall" references create structural traps during preference optimization is conceptually neat and supported by clear empirical evidence.
- The experimental setup is simple yet effective. The Mixture-of-Gaussians example isolates the mechanism cleanly, and the follow-up experiments on the SmolLM2 family demonstrate that the phenomenon persists at scale.

**Weaknesses:**

Having agreed upon the main message, my major concerns lie in the quality of exposition. Overall, the writing is often too rhetorical rather than mathematically precise. Key terms and concepts are used before being properly introduced or defined, which makes the argument difficult to follow. I also have several suggestions regarding the writing style listed below. I strongly believe that the paper requires a major revision to improve clarity and readability. With such a revision, the paper could become a strong and useful reference for the community. Detailed comments are provided below.

- The writing style of the paper is not linear. The exposition often moves back and forth between motivation, intuition, and definition instead of following a clear logical order. As a result, it is difficult for the reader to understand the conceptual framework before being asked to interpret technical claims. The manuscript tends to introduce important terms long before defining them and then revisits those terms multiple times in later sections, each time with slightly different or incomplete explanations. Here are some examples.

    - For instance, the term **"distributional recall"** appears in the abstract and introduction as if it were already a well-defined concept. However, no explicit mathematical definition or formal description is given at that point. The reader is left to guess what "recall" means in this probabilistic context until Section 3, where the authors begin to use it operationally. Even there, the explanation is scattered across paragraphs and depends heavily on the reader's prior familiarity with KL regularization.

    - Similarly, the notion of **"desirable behaviors"** is introduced early in the paper in the phrase *"we posit a simple recall requirement: desirable behaviors must lie within the support of π_ref"*. This statement is highly ambiguous because "desirable behaviors" could refer to several different things.

    - The authors also use the term **"low-recall trap"** repeatedly but define it gradually in several fragments rather than once in a clear and complete form. The paper first mentions it in the introduction as a structural problem, later splits it into a "sampling trap" and "learning trap," and only then provides some mathematical reasoning behind it. The lack of a single, unified definition forces the reader to reconstruct the meaning by piecing together several paragraphs. This pattern makes the presentation feel recursive rather than progressive.

The cumulative effect is that the paper's theoretical argument feels less rigorous than it actually is. The ideas themselves are coherent, but the order of presentation obscures them. To improve readability, the authors should define all key terms explicitly and early. A linear presentation would begin by listing the essential concepts (for example, what "recall" means for a model distribution, what counts as a "desirable behavior," and what exactly constitutes the "low-recall trap"), followed by clear empirical or mathematical demonstrations. Without such structure, the argument appears handwavy and repetitive even when the underlying logic is sound.

**Questions:**

- It seems that the critical (yet implicit) assumption is that the "distillation" procedure yields a low-recall model. I suspect that it is based on (Cha and Cho, 2025), but it seems to be only implicitly assumed. I suggest the authors to add an explicit discussion on the assumption, and some justification if possible.
- Why Section 4.1 and Section 4.2 are not under a single subsection?

#### **Suggestions**
- Besides the writing, the manuscript uses italics and bold faces too frequently for emphasis. I recommend limiting typographic emphasis to key definitions or novel terms only, as excessive use can distract the reader and reduce the perceived objectivity of the presentation.
- Inconsistent formatting in section headings. Remove periods in the headings of Sections 3.1 and 3.2, for example.
- In Section 4.1, please explicitly mention that the dimension is two. Overall, the description seems incomplete.
- I found the oracle reward function $R(x)$ in Appendix A, but this should be explicitly mentioned.
- Consider indexing $p''$, instead of using the single notation $p''$ for two distributions (4 modes / 3 modes).
- "Decisive" in line 113 does not sound idiomatic. I suggest to revise the manuscript carefully.
- In line 235, remove "in" before "real LLM experiments".
- Please go over the references and fix capitalization.

---

> ### Author Response · Authors · 2025-11-17
>
> We thank you for your time and effort in reviewing our paper. We appreciate the constructive feedback on our writing style and for acknowledging the importance of this work.
>
>
> ## Weaknesses:
>
> ### 1. Concern regarding the writing style and the order of presentation
>
> Thank you for these constructive and detailed comments. We have reviewed our manuscript against your feedback and are in **complete agreement** that the current version suffers from the readability and clarity issues you identified. The "non-linear" presentation obscures the rigor of our claims, and we have already begun work on a major revision to fix this.
>
> We will first outline our improvement plan below. We will then upload the fully revised paper and a summary of changes as soon as possible during the discussion period.
>
> > The term 'distributional recall', 'Desirable behaviors', and 'low-recall trap'.
>
> We sincerly thank the reviewer who raises this issue. In our revision, we will address this in two ways:
>
> 1. We will provide clear, formal definitions for key terms, specifically "distributional recall" (citing relevant literature like [*Why Knowledge Distillation Works in Generative Models: A Minimal Working Explanation, Cha & Cho, NeurIPS 2025*]), "desirable behaviors" (defining them as rare, high-reward modes), and the "low-recall trap."
> 2. We will revise the Abstract and Introduction to describe these concepts in detail first, rather than using the terms before they are formally defined. This will ensure a smoother and more accessible introduction for the reader.
>
> > The writing is often too rhetorical rather than mathematically precise
>
> We agree. As part of our revision, we will meticulously review the entire manuscript to replace rhetorical phrasing with more precise and technical language to support our claims.
>
> ## Questions
>
> ### 1. It seems that the critical (yet implicit) assumption is that the "distillation" procedure yields a low-recall model... I suggest the authors to add an explicit discussion on the assumption...
>
> Thank you for the reviewer's question. This assumption is indeed based on the findings in [*Why Knowledge Distillation Works in Generative Models: A Minimal Working Explanation, Cha & Cho, NeurIPS 2025*]. We will add an explicit discussion on this assumption in our revised section on the "low-recall trap." We will provide this justification and also point to our own empirical results (e.g., Fig. 2) which validate this assumption in our setup.
>
> ### 2. Why Section 4.1 and Section 4.2 are not under a single subsection?
>
> This is a valid structural point. In the revised version, we will combine Sections 4.1 and 4.2 into a single, cohesive section (e.g., "4.1 Mixture-of-Gaussians Validation") to improve the paper's readability and logical flow.
>
> ## Suggestions
>
> We thank the reviewer for these helpful suggestions.  We will address all 8 of these points in our revision. We will reduce typographic emphasis, fix all formatting inconsistencies, add the missing experimental details (dimension, reward function), update the notation for p′′, and perform a careful copy-edit of the entire manuscript to fix word choice and references.
>
> ---
>
> We thank the reviewer again for their highly constructive feedback. As outlined above, we are first submitting this response detailing our action plan. The revision is already in progress. We are working to complete it and will upload the revised manuscript, along with a summary of the detailed changes, as soon as possible during the discussion period. If you have additional concerns or questions, please post it anytime.

---

> ### Author Response · Authors · 2025-11-21
> **Author response after revision**
>
> We sincerely thank you for the constructive feedback. We have focused extensively on improving the overall readability by addressing the "non-linear structure" you identified as a major weakness. Below is a detailed summary of the revisions made to address your concerns:
>
> * **Clarification of Key Terms (‘Distributional recall’, ‘desirable behaviors’, ‘low-recall trap’):**
>     We minimized the use of these technical terms in the Abstract and Introduction, replacing them with more intuitive, general descriptions to improve readability. Specifically, we added a paragraph at the end of the Introduction to conceptually introduce the "low-recall trap" and its components ("sampling trap" and "learning trap"). Formal definitions and mathematical notations for these terms are now systematically presented in Section 3.4 to ensure a linear logical flow.
>
> * **Explanation of the Premise ("Distillation yields a low-recall model"):**
>     We revised Sections 3.3 and 3.4 to address this point explicitly. Section 3.3 now focuses solely on the current practice of using distilled models as reference models. In Section 3.4, we added a new paragraph titled "**Premise: Distillation Reduces Distributional Recall**." In this section, we explicitly discuss the precision-recall trade-off inherent in Knowledge Distillation (Cha & Cho, 2025) and provide formal definitions for "low-recall reference models" and "desirable behaviors." This establishes a clear logical foundation for the subsequent explanations of the Sampling and Learning Traps.
>
> * **Restructuring Sections 4.1 and 4.2:**
>     We merged the original Section 4.1 and Section 4.2 into a single, cohesive section titled **Section 4.1: "Empirical Validation with Mixture-of-Gaussians."**
>
> * **Typographic Emphasis (Italics and Boldface):**
>     We have significantly reduced the use of bold and italic text throughout the manuscript, limiting typographic emphasis to key definitions or essential points only.
>
> * **Formatting Consistency:**
>     We standardized the formatting of section headings throughout the paper, including corrections to the inconsistencies noted in Sections 3.1 and 3.2.
>
> * **Explanation of the Oracle Reward Function:**
>     * For the MoG experiments, we added a dedicated "**Oracle Reward Function**" paragraph in Section 4.1 with a detailed explanation.
>     * For the LLM experiments (Section 4.2), we separated the reward description into a dedicated "**Oracle Reward Function**" paragraph. Since the formulation for LLMs is more complex, we provided the full details in the Appendix and added explicit reference in the manuscript.
>
> * **Indexing Notation and Dimensionality:**
>     Following your suggestion, we introduced distinct notations, $p''_3$ and $p''_4$, to represent the 3-mode and 4-mode KD models, respectively (See Lines 262-264), and used these consistently to describe the MoG results. We also explicitly stated that the MoG experiment is conducted in a **2D space** in Section 4 (See Lines 257-263).
>
> * **Language and Grammar:**
>     We carefully reviewed the manuscript and revised awkward expressions and grammatical errors to ensure a natural and professional tone.
>
> * **References:**
>     We audited the bibliography and corrected the capitalization for proper nouns and acronyms, ensuring consistency for terms such as SmolLM2, PaLM 2, PyTorch, DistilBERT, LLM, etc.
>
> We believe these revisions have substantially improved the manuscript's readability and structural coherence. We are grateful for your insightful comments. We remain committed to further polishing the text for clarity in the final camera-ready version.

---

### Author Response · Authors · 2025-11-17
**Common response 1 for Reviewer RXEC and v1RN**

## [Common response 1 for Reviewer RXEC and v1RN: Experimental setup of LLM experiments]


In practice of LLM KD/alignment research, we only consider to use a teacher (p'; the reference model) and a student (p''; the aligned model). This "real-world" scenario makes it impossible to measure a model's general capability against the true ground-truth distribution (which is unknown). To evaluate the student model trained by KD, we must rely on downstream benchmark tasks, which only evaluate specific capabilities, not the model's overall distributional quality (i.e., its general precision and recall).

To overcome this, we adopted the methodology from [*Why Knowledge Distillation Works in Generative Models: A Minimal Working Explanation, Cha & Cho, NeurIPS 2025*]. This required three models to create a controlled setup where we can measure precision and recall against the GT distribution. The table below clarifies the roles:


|Case | GT (p*) | Pretrain (p') | Target model (p'') |
| -------- | -------- | -------- | -------- |
| Real-world | Real-world Text Dist. (unknown)     | Pretrain p' with sampled data from p* | Train p'' with KD->Align or Align->KD using p'     |
| Our setting | Pretrained SmolLM2 1.7B | Pretrain p' (SmolLM2 360M) with sampled data from p* | Train p'' (135M) with KD->Align or Align->KD using p'

In our setting, the pretrained SmolLM2 1.7B model acts as the known GT text distribution (p∗). This allows us to:

1. Faithfully simulate the real-world process: We train p′(360M) from p∗, and p′′(135M) from p′.
2. Perform a more rigorous evaluation: Because we know p∗, we can calculate the true precision and recall of our final model (p′′).


As discussed in Lines 310-318 of the manuscript, note that precision (E_x∼p''[logp∗(x)]) measures how well the final model p′′ is aligned with the target modes within the GT distribution. Recall (E_x∼p∗[logp''(x)]) measures how well p′'' covers the entire GT distribution.


This setup and metrics allow us to prove, in a general and comprehensive way, that **Align $\rightarrow$ KD consistently produces a final p′′ that is both better aligned (higher precision) and retains more general knowledge (competitive recall) than the KD $\rightarrow$ Align pipeline**.

---

### Author Response · Authors · 2025-11-17
**Common response for all reviewers**

## [Common response for all reviewers]

We sincerely thank all reviewers for their constructive feedback. We are encouraged by the consensus on our work's core strengths: namely, that we address an "important" (Reviewer aqcD) and "timely" (Reviewer XYEC) question in LLM post-training, supported by "clear empirical evidence" (Reviewer aqcD and XYEC) from a "robust" (Reviewer XYEC) experimental setup that "isolates the mechanism cleanly" (Reviewer aqcD). We wish to clarify that our primary contribution is not simply the intuition that "high-recall models align better," but rather, as Reviewer aqcD noted, **the "formalization" of why the common, efficiency-driven KD $\rightarrow$ Align pipeline is structurally flawed**. We identify this failure as the "low-recall trap" , a specific mathematical barrier in PPO and DPO objectives  that prevents the recovery of rare modes. **Our work validates Align $\rightarrow$ KD as the robust solution to this trap.** We are addressing all specific feedback, especially on presentation (Reviewer aqcD) and narrative clarity (Reviewer XYEC), and will upload a revised manuscript soon.

---

### Author Response · Authors · 2025-11-21
**Common response for all reviewers (after revision)**

## Common response for all reviewers (after revision)

We have completed the revision process following our initial author responses and have uploaded the revised manuscript along with a summary of the changes. Our primary objective for this revision was to prioritize and effectively address the major weaknesses identified by each reviewer, while keeping the changes focused and concise.

For detailed explanations of specific revisions, please refer to our individual responses to each reviewer. To facilitate your review, all updates in the revised manuscript have been **highlighted in** **purple**.

We would like to express our sincere gratitude once again for your insightful and constructive comments. We are fully committed to incorporating all remaining feedback in the final camera-ready version to further enhance the quality and clarity of our work. Thank you.

---

### Meta-Review · Area_Chair_tFfo · 2025-12-30

**Summary:**

This paper challenges the common "distill-then-align" (KD $\rightarrow$ Align) pipeline, arguing it leads to a "low-recall trap" where the distilled model loses rare but desirable modes that cannot be recovered. It proposes an "align-then-distill" (Align $\rightarrow$ KD) approach, supported by Mixture-of-Gaussians (MoG) and LLM (SmolLM2) experiments.

The reviewers acknowledge the importance of the problem and the clear intuition behind the proposed "low-recall trap" mechanism. The empirical validation on MoG is seen as a strong proof of concept. However, the reviewers raise several core concerns including (1) Poor presentation and writing quality (non-linear structure, recursive definitions), (2) Inconsistencies in the narrative (conflating reference model quality with pipeline order), and (3) The realism of the experimental setup (using a 3-model proxy to simulate ground truth vs. standard industrial practices).

While the authors acknowledged the presentation issues and provided clarifications on the experimental design in their rebuttal, the fundamental structural issues in the manuscript remain unresolved.

In summary, given the severe issues with the paper's organization and the outstanding concerns regarding the applicability of the experimental setup to real-world scenarios, I recommend the submission for Reject.

While this is indeed an insightful paper, I encourage the authors to restructure the narrative and definitions linearly. Crucially, I suggest the authors further validate their claims in a more standard setting that closely matches real-world industrial pipelines for future resubmission. Aligning the experiments with industrial deployment scenarios will significantly enhance the persuasiveness and impact of the work.

**Reviewer Concerns:**

Addressed Concerns:

- The authors substantially improved the paper’s clarity, structure, and technical rigor by reorganizing the narrative into a coherent, linear progression and resolving major readability issues (aqcD).
- Previously implicit assumptions—most notably that knowledge distillation induces a low-recall model—were made explicit and justified through prior literature and new empirical evidence (aqcD, v1RN).
- Core concepts such as distributional recall, desirable behaviors, and the low-recall trap were clearly defined and formalized, replacing rhetorical phrasing with precise technical language (aqcD, NvRw).
- Conceptual confusion between SFT, KD, and alignment was resolved through explicit definitions and clearly specified training pipelines (XYEC).
- Empirically, the authors provided consistent evidence across controlled Mixture-of-Gaussians experiments, language-model experiments, and multi-mode settings, supporting the claim that the Align → KD ordering avoids a structural loss of recall and improves robustness (XYEC, v1RN).
- Technical issues related to notation, oracle reward specification, section organization, and experimental details were systematically addressed (aqcD).



Despite these improvements, several high-level concerns remain only partially resolved:

- The external validity of the findings is limited, as experiments are confined to small models and controlled settings, with no evaluation on large-scale benchmarks or complex real-world alignment tasks (v1RN).
- Computational cost trade-offs—central to the practical motivation for the KD → Align paradigm—are acknowledged but not quantitatively analyzed, leaving the real-world cost–benefit balance unclear (v1RN).
- Alternative mitigation strategies within the standard KD → Align pipeline, such as enhanced distillation or diversity-preserving techniques, are not systematically explored (v1RN).
- In addition, broader questions regarding practical relevance—particularly whether alignment inherently reduces diversity or recall, and how distributional recall should map onto real-world alignment objectives—remain open and partially unresolved (XYEC, NvRw).

**Reviewer Scores:**

- Reviewer aqcD: 4 -> 4

This reviewer’s evaluation was dominated by concerns about clarity, structure, and conceptual precision rather than technical correctness. Although the authors explicitly acknowledged these weaknesses and outlined a comprehensive plan to restructure the paper—introducing formal definitions, linearizing the narrative, and reducing rhetorical language—the reviewer’s stated acceptance criterion hinged on verifying a major revision to presentation quality. Because such large-scale structural improvements cannot be fully assessed during the discussion phase, the reviewer has insufficient evidence that the readability and coherence issues have been resolved in the current manuscript. As a result, despite strong agreement on the paper’s technical soundness and importance, the score remains unchanged and marginally below the acceptance threshold.

- Reviewer XYEC: 2 -> 4

The reviewer initially expressed skepticism about the novelty and focus of the contribution, particularly whether the paper merely restated a well-known intuition—that high-recall reference models are preferable for alignment—or provided a distinct, mechanism-level insight. In response, the authors clarified the paper’s central claim by disentangling the requirement for a high-recall reference model from the procedural question of alignment–distillation ordering, and by formalizing the “low-recall trap” as a structural failure mode rather than a heuristic observation. They also resolved prior confusion surrounding SFT, KD, and alignment by explicitly defining each stage and presenting concrete pipeline formulations. While broader questions about real-world relevance and normative alignment objectives remain open, the core conceptual ambiguity was addressed sufficiently to justify a substantial upward revision of the score to a borderline-accept range.

- Reviewer v1RN: 4 -> 6

This reviewer’s concerns centered on the methodological validity and interpretability of the experimental setup rather than the core idea itself. The authors convincingly clarified that their three-model formulation and controlled experimental design are intended to isolate precision–recall dynamics in post-training pipelines, rather than to serve as a direct recipe for deployment. By consistently supporting the “low-recall trap” mechanism across Mixture-of-Gaussians, language-model, and multi-mode experiments, and by explicitly framing the contribution as a mechanism-level validation rather than a benchmark claim, the authors addressed the reviewer’s primary doubts about soundness. Although concerns about scalability, computational cost, and external validity remain unresolved, the reviewer appears satisfied that the central claims are internally consistent and scientifically justified, supporting a moderate score increase.

- Reviewer NvRw: 4 -> 4

This reviewer’s assessment is primarily driven by concerns about empirical coverage and real-world relevance. While the authors strengthened the theoretical motivation of their approach and clarified the mechanism-level interpretation of their results, the rebuttal does not materially expand the experimental scope toward realistic deployment settings. In particular, the absence of evaluations on standard large-scale benchmarks or industrially representative alignment tasks leaves the reviewer’s doubts about practical significance unresolved. Because this reviewer places decisive weight on empirical validation in real-world scenarios, the rebuttal’s emphasis on conceptual justification rather than closing the experimental gap is unlikely to warrant a score increase.

---

### Decision · Program_Chairs · 2026-01-26

Reject